



# A data-centric perspective on the information needed for hydrological uncertainty predictions

Andreas Auer[1], Martin Gauch[2], Frederik Kratzert[3], Grey Nearing[4], Sepp Hochreiter[1], and Daniel Klotz[5]

[1]ELLIS Unit Linz and LIT AI Lab, Institute for Machine Learning, Johannes Kepler University Linz, Austria
[2]Google Research, Zurich, Switzerland
[3]Google Research, Vienna, Austria
[4]Google Research, Mountain View, California, USA
[5]Department of Compound Environmental Risks, Helmholtz Centre for Environmental Research — UFZ, Leipzig, Germany

**Correspondence:** Andreas Auer (auer@ml.jku.at)

**Abstract.** Uncertainty estimates are fundamental to assess the reliability of predictive models in hydrology. We use the framework of Conformal Prediction to investigate the impact of temporal and spatial information on uncertainty estimates within hydrological predictions. Integrating recent information significantly enhances overall uncertainty predictions, even with substantial gaps between updates. While local information yields good results on average, it proves insufficient for peak flow

predictions. Incorporating global information improves the accuracy of peak flow bounds, corroborating findings from related studies. Overall, the study underscores the importance of continuous data updates and the integration of global information for robust and efficient uncertainty estimation.

## 1 Introduction

Uncertainty estimates are the basis for actionable predictions (e.g., Krzysztofowicz, 2001; Beven, 2016a). In this contribution we take a data-centric perspective to investigate how temporal (recency) and spatial (local vs. global) information impacts the quality of uncertainty estimates. With machine learning, we investigate on the level of information content with respect to different tasks, rather than on the level of process representations. For our study, we utilize Conformal Prediction (CP, Vovk et al. (2005)), a model agnostic framework that adds uncertainty intervals to existing predictions. In contrast to many ad-hoc

approaches for uncertainty estimation, CP is motivated in a rigorous, probabilistic manner. Under the right conditions, CP intervals will always achieve finite-sample marginal coverage (Vovk et al., 2005). Further, the model agnostic nature of CP enables the separation of a model's point prediction quality from its uncertainty prediction quality. This is, for example, not possible with the Deep Learning baselines from Klotz et al. (2022), which serve as a reference wherever adequate. Specifically, we apply HopCPT (Auer et al., 2023), the state-of-the-art CP model for time series. HopCPT memory-based architecture

determines which data points drive the uncertainty prediction during inference. The identification of data points that are relevant




for uncertainty, can be exploited to analyze the impact of selected information on the predictive outcomes in an indirect yet flexible way.

To our knowledge, we are the first to introduce CP to hydrology. Our goal is to use CP as a tool to empirically study the following research questions (RQ):

– **RQI:** *Does up to date information improve uncertainty estimation?*
We show that continual updates greatly benefit the general uncertainty prediction, even if the updating happens in batches (i.e. discontinuously).

– **RQII:** *Is the data from a given basin required to get good uncertainty estimates for that basin?*
Our results suggest that in the general case local information is indeed beneficial for uncertainty estimation, since it will
generally result in tighter prediction intervals. In a PUB setting, while larger intervals are necessary, it's still possible to achieve quite good performance in terms of uncertainty prediction.

– **RQIII:** *Is data from a single basin enough to get good uncertainty predictions for peak flows?*
Our results indicate that it is necessary to use data from different basins to provide good uncertainty predictions for peak flows at a given basin.

**Trading space for time in hydrological modeling.** Multiscale Parameter Regionalization (MPR; Samaniego et al., 2017; Schweppe et al., 2022) is a technique that allows to calibrate hydrological models using a scale independent, global parametrization. MPR thus uses all available data to provide an estimation for the parameters of a given basin (the possible parametrization is, however, still strongly restricted by a priori knowledge in the form of the model structure and the functions that link the spatial information to the model parameters). Similarly, Kratzert et al. (2019b) introduced a Long Short-Term Memory (LSTM)
rainfall-runoff model that is globally parametrized. In terms of predictive fidelity, this LSTM based approach outperformed many classical rainfall–runoff models (e.g., Kratzert et al., 2019a, 2021; Mai et al., 2022). Klotz et al. (2022) showed how this LSTM based approach can directly provide predictive uncertainty estimations. An inspection of the importance of data for the LSTM based approach can be found in Gauch et al. (2021). They concluded that adding multiple basins (i.e., the spatial part of the data) is key for reaching good model performances. In time series prediction in general, Montero-Manso and Hyndman
(2021) found that global modelling approaches — such as the one discussed here — tend to outperform local ones. A different research direction with similar implication is the contribution by Bertola et al. (2023), who analyzed how floods from different regions are informative of each other. They show that many observed floods fall within the envelope values estimated from previous floods in other basins. This suggests that local flood predictions can benefit from information from different places. We are not aware of any publications that explicitly examine the space and time relationship for uncertainty estimations.


**Uncertainty estimation in hydrology.** Uncertainty has long been recognized as a crucial part of hydrological modelling (e.g., Krzysztofowicz, 2001; Beven, 2016b). Thus, there already exists a wide range of approaches for uncertainty estimation in hydrological modeling. As of today, approaches include — but are not limited to — ensemble-based methods, that define and





sample probability distributions around different model inputs, structures, or outputs (e.g., Li et al., 2017; Demargne et al.,
2014; Clark et al., 2016); Bayesian and Bayesian inspired techniques, which weight different parameters, models, or outcomes (e.g., Kavetski et al., 2006; Beven and Binley, 2014); neural network based methods which estimate the parameters (of mixtures) of probability distributions (Klotz et al., 2022, e.g.); and even explicit post-processing methods (e.g., Shrestha and Solomatine, 2008; Montanari and Koutsoyiannis, 2012; Koutsoyiannis and Montanari, 2022). Good overviews can for example be found in Nearing et al. (2016) or Gupta and Govindaraju (2023).

**CP for time series.** The current state of the art in conformal time series prediction is HopCPT, a CP approach based on deep learning (Auer et al., 2023). To provide a prediction interval for a given basin and time step, HopCPT learns to retrieve historical time steps that belong to similar regimes — i.e., time steps that had similar error patterns. Intuitively, the CP model performs a soft nearest-neighbor search with a learned similarity measure. This leads to tighter and more accurate uncertainty estimates than existing approaches, as it incorporates knowledge not just about the marginal distribution but also about the current system state. HopCPT's regime definition, which considers regime changes within a time series, is loosely related to regimes in the hydrological modeling sense (Haines et al., 1988; Harris et al., 2000), which classify rivers according to the overall flow behavior. The definitions of Quandt (1958) and Hamilton (1990), which model time series with multiple regimes where the distribution parameters are conditional on the active regime, are closer to our actual use of the regime term.

The remainder of this paper is structured as follows: First, Section 2.1 provides an introduction to CP geared towards hydrologists and time series prediction. Section 2.2 describes the methods relevant to this study (HopCPT, CMAL), and Section 2.3 describes the metrics used for comparing the different approaches. In Section 3 we present our experiments and corresponding data. Section 4 details and discusses the results. Finally, we conclude in Section 5.

## 2 Methods

### 2.1 Conformal Prediction

This section provides a brief overview of CP. For a thorough introduction to Conformal Prediction, we refer the reader to Angelopoulos and Bates (2021).

A CP procedure consists of two steps: First, CP estimates the "unusualness", here called non-conformity, of data points within a calibration set — which contains previous hold-out data that are not used for training the prediction model. Then, CP uses this information directly to construct an uncertainty region that consists of the most "usual" values of the calibration set.

There does not exist a definitive function to measure non-conformity. As a matter of fact, there are infinitely many non-conformity measures. Even a function that randomly assigns a value of an arbitrary distribution would allow the coverage guarantee of CP to hold — as long as the distribution does not change between calibration and test samples. However, choosing a non-conformity measure that yields good prediction intervals — in the sense that they are not too broad — is part of the challenge when applying CP. Vovk et al. (2005), for instance, point out that whether any particular approach is an appropriate



way to measure non-conformity depends greatly on contextual factors. In a regression setting a straight-forward example for a non-conformity measure is the absolute error of the prediction.

Even the most basic CP approaches do not pose specific assumptions to the underlying data distribution (for comparison,
CMAL, as proposed by Klotz et al. (2022), assumes a mixture of asymmetric Laplacians) except that the data are exchangeable (i.e., the joint distribution is invariant to permutations of the data). Hydrological tasks — such as streamflow prediction — typically violate this exchangeability assumption, since errors are highly correlate in time and exhibit different behavior for different situations — e.g., a model might have much larger errors in a flood situation than in a low-flow situation. On top of that, environmental processes, especially when considering long time periods, likely exhibit shifts in the data distribution.
These can arise from a spectrum of factors, ranging from gradual changes, such as those induced by climate change, to more accelerated transformations, like those stemming from infrastructure projects. Since the prediction interval of CP is based on the calibration set, i.e., based on past observations, such shifts can lead to unreliable prediction intervals. Formally, we can also view this as a break from the exchangeability assumption. Besides that, standard CP generates prediction intervals that provide marginal coverage, which produces unnecessarily wide or too small prediction intervals when different error patterns exist. For
example, in time series regression, given the absolute error as a simple non-conformity score, the prediction interval would have the same width over the whole time sequence. Recent advances in CP methods have tackled these inherent problems and thus made it possible to use CP for time series uncertainty estimation. Typically, they either adapt the calibration distribution based on the temporal proximity and/or the time series covariates (Auer et al., 2023; Foygel Barber et al., 2022; Xu and Xie, 2022a, b). Other variants propose an adaptive prediction interval (Gibbs and Candes, 2021; Zaffran et al., 2022; Bhatnagar
et al., 2023) that operates in an online fashion and therefore needs access to the label measurements after prediction.

To be more explicit, we consider a setting where we have a calibration dataset $D = \{(x_1; y_2)(x_2, y_2), \ldots, (x_n, y_n)\}$. Applying a prediction model $\mu : X \mapsto Y$ gives us the (absolute) errors of the calibration data $E = \{e_1, e_2, ..e_n\}$ — these errors represent the non-conformity scores as a higher error refers to a more "unusual" sample (Figure 1-left). Our goal is to create a prediction interval for a new sample $(x_{n+1}; y_{n+1})$ which covers the unknown error $e_{n+1}$ with probability $1 - \alpha$, where $\alpha$ represents the
specified miscoverage rate. The standard CP procedure is as follows: (1) We use the $1 - \alpha$ empirical quantile of $E$ to estimate the score for which a $1 - \alpha$ ratio of the samples have a lower score — i.e., are less "unsual" (Figure 1-middle). (2) Because we assume the data is exchangeable, the non-conformity score of the new sample is represented by the same distribution. Therefore we can simply define the prediction by adding and subtracting the quantile score to the model prediction to arrive at the lower and upper bound respectively (Figure 1-right).

In a classical CP setting it is important that the calibration set is not part of the training data of the model $\mu$. This is because the fit on the training data is biased: models can, for example, overfit. Therefore, the non-conformity score distribution of the training data will likely not generalize to new data. Also, many CP approaches assume that the model itself is already capable of providing uncertainty estimates. This is however not a necessity. Conformal prediction can be used in classification or regression settings for point, interval, and distributional predictions.

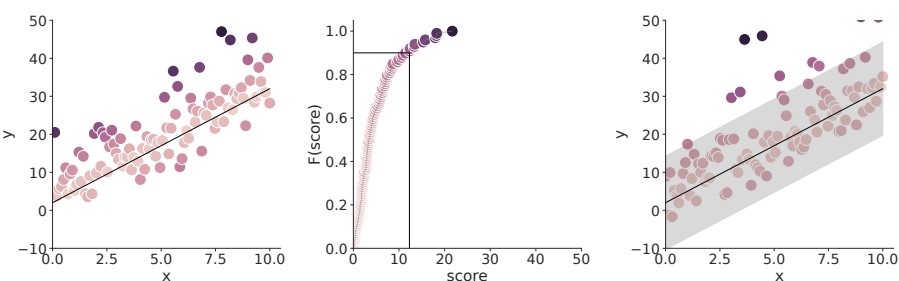

**Figure 1.** Illustration of standard conformal prediction applied to a regression setting. Left: The black line shows the prediction of a model. The colored points show the real observation — the darker the color the bigger the prediction residual, i.e., the non-conformity score. Middle: CDF of the non-conformity score distribution and the respective cut-off quantile at $\alpha = 0.1$. Right: The prediction of $\mu$, with the CP interval defined by the value on the cut-off quantile, on the test data.

## 2.2 Uncertainty Estimation Models

This section introduces the models and metrics used to answer our research questions. We introduce HopCPT and its "global" variant, HopCPT-G, in Section 2.2.1 and Section 2.2.2 respectively. Section 2.2.3 presents CMAL.

### 2.2.1 HopCPT

Why are modeling systems often successfully used for decision making even if they do not provide uncertainty predictions? We believe that this is because decision makers implicitly consider the model behavior over time and compare a given forecast to the recent performance and the model behavior in similar situations. We thus design HopCPT (Auer et al., 2023) to explicitly capture this notion in a quantitative way.

HopCPT is the current state-of-the-art approach for conformal time series prediction. The uncertainty of time series data often varies heavily between certain periods. One reason are seasonal patterns. For example, for many Alpine rivers, long-lasting periods of low-flows in winter exhibit lower predictive uncertainty than large events in late spring where glacier melt and convectional precipitation events interact. Another reason is the occurrence of irregular but recurring events such as those induced by torrential rain. Note that shifts play an important role also within events, as their frequency and intensity can change over time. HopCPT addresses these challenges by viewing the time series as a soft partition of time periods, where each partition element exhibits individual uncertainty properties. We refer to such a set of time points as a regime (Quandt, 1958; Hamilton, 1990) and assume that one can identify them by the covariates and the lagged target of the time series. HopCPT learns to weight which past observations — i.e., calibration points — are likely from the same regime as the current point[1]. Based on this information, the different calibration points are weighted differently when constructing the prediction interval. Since the considered time steps in the interval calculation are from the same regime, one can assume that exchangeability is

---

[1]We want to emphasize that calibration data refers here — and throughout the manuscript — to data that is not utilized in training the underlying point prediction model. This aligns with the conventional terminology in probabilistic applications in general and the CP literature in specific.





given and therefore the validity of the prediction interval holds. Yet, since time points from unrelated regimes are disregarded,
HopCPT results in tighter intervals than, for example, standard CP.

More formally, HopCPT assigns a weight $a_{t,i}$ to a past time step $i \in M$ in the memory $M$ given the current time step $t$. The weight reflects the similarity between the given time step $i$ and the current time step $t$. Intuitively we can say that the weight should be high when $i$ is from the same regime as $t$, and otherwise low. These weights are constructed with the Modern Hopfield Network component of HopCPT as follows

$$a_{t,i} = \beta\, m(\boldsymbol{z}_t)\, \boldsymbol{W}_q\, \boldsymbol{W}_k\, m(\boldsymbol{z}_i), \tag{1}$$

$$\bar{a}_{t,i} = \frac{e^{a_{t,i}}}{\sum_{j \in M} e^{a_{t,j}}} \quad \forall i \in M, \tag{2}$$

where $\boldsymbol{z}_i$ is the representation input for time step $i$, $m$ represent an encoding module, and $\boldsymbol{W}_q$ and $\boldsymbol{W}_k$ are learned weight matrices; $\beta$ is a hyperparameter that that represents the inverse of the so-called softmax temperature, which adjusts the sharpness of softmax-generated probability distributions (low temperatures push all the attention to a single value; while high temperatures
distribute the attention uniformly).

The weights form the basis for the weighted conformal prediction interval (Foygel Barber et al., 2022). Besides the weighting, HopCPT deviates from standard CP as it does not consider the one-sided quantile of the absolute errors (non-conformity scores), but instead follows Xu and Xie (2022a) and excludes the lower and upper quantiles of the relative errors. Formally,[2] the prediction interval of time step $t$ is calculated by

$$\widehat{C}^{\alpha}(\boldsymbol{z}_t, M) = \left[ \hat{\mu}(\boldsymbol{X}_t) + \mathsf{Q}_{\frac{\alpha}{2}}\left( \sum_{i \in M} \bar{a}_{t,i}\delta_{\epsilon_i} \right),\, \hat{\mu}(\boldsymbol{X}_t) + \mathsf{Q}_{1-\frac{\alpha}{2}}\left( \sum_{i \in M} \bar{a}_{t,i}\delta_{\epsilon_i} \right) \right], \tag{3}$$

where $\mathsf{Q}_\tau$ is the $\tau$-quantile and $\delta_{\epsilon_\tau}$ is a point mass distribution (i.e., a degenerate distribution where all the mass is concenrated at a single point) at the prediction error at time step $i$. This essentially means we compute the $\tau$-quantile across a mixture of $M$ distributions, each corresponding to a distinct time step $i$ in the memory — characterized by a point mass at its prediction error $\epsilon_i$. Thus, each of these distributions is concentrated at $\epsilon_i$ with density 1 and 0 elsewhere.

HopCPT retrieves the calibration data points during the prediction from the Hopfield Memory (Ramsauer et al., 2021) of the model. Thus, one can simply add every new and available observation to the memory. This corresponds to an automatic recalibration, which accounts for shifts in the data distribution. In addition, HopCPT allows to add a so-called temporal encoding to the time steps. This encoding adds information about the time difference between the predicted time step and the previous time steps. Given this information, HopCPT can learn to weight recent points higher, which further helps to address distribution
shifts.

HopCPT has already exhibited good performance in streamflow uncertainty prediction (Auer et al., 2023). In this work we use HopCPT approach as a tool to explore the effects of data availability on uncertainty prediction.

**Memory update.** In Auer et al. (2023), HopCPT updates the Hopfield Memory after each prediction with the — then —
previous prediction error. However, this requires access to the target label of this time step to calculate the error of the prediction

---

[2]Note that we use the alternative proposal presented in Appendix E of the original work as it is computationally more effcent



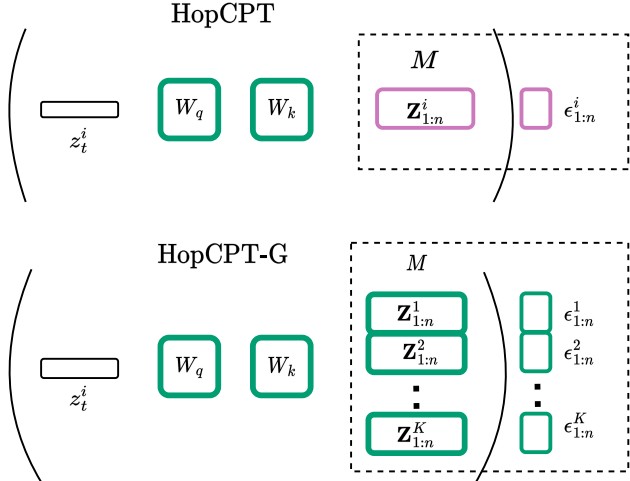

**Figure 2.** Illustration of the difference between HopCPT and HopCPT-G. The shared weights and vectors are outlined in green, the vectors individual to a specific time series in magenta. HopCPT-G also shares the memory vectors and incorporates all available time steps.

model. In streamflow modeling, we often do not have direct access to the target label (i.e., the streamflow measurement) before a new prediction has to be issued. Therefore, we adapt HopCPT to use a fixed memory that is only based on the calibration data by default. We refer to this adaptation as the "offline mode", in contrast to the original "online mode". Given the notation from Equation 3 this means that in the online mode the memory $M$ contains all time steps up to time step $t-1$, while in the

offline mode, the memory is fixed per time series and only includes the calibration data.

In practice, one could expect that the memory may be updated intermittently, whenever new data becomes available. Hence, we explore the influence of different updating schemes (Experiment I).

**Input features.** Auer et al. (2023) concatenate (static and dynamic) time series covariates, model predictions, and lagged

targets as input features to generate the representation $z_i$ of a time step $i$. However, the use of the lagged target requires access to the target label right after the prediction, which is typically not the case in streamflow prediction. A straightforward solution would be to simply exclude the lagged target as an input feature. However, since most current rainfall–runoff models follow a state-space approach, we also explore how lack of labels can be compensated by using the model state. We hypothesize that this state should include most of the relevant information that would have been provided by the streamflow observations, as well

as additional information that is not available to HopCPT in the original publication. Note that (a) the model states potentially contain more information than just the model prediction because the prediction is some projection of its state[3] and (2) the state likely encodes important historical information beyond the current time step.

---

[3]Here we assume that the model state at time $t$ already considers the input features of time step $t$.





### 2.2.2 HopCPT-G

We hypothesize that the union of the existing calibration data represents not only the included streamflow time series, but rather
a general set of existing streamflow regimes. In this scenario one could utilize the error information from similar situations
in other basins to best model the current situation in a specific basin. Further, the application to ungauged basins should be
possible. In a simple case, where the error information is only relevant within a certain time series, HopCPT can learn to fall
back to the local setting and only up-weight the calibration data of its own time series. To examine the potential of global
memory, we modify HopCPT so that it operates on a global memory for both training and inference. We refer to this new
model as HopCPT-G.

During inference, the association weights and prediction interval calculation of HopCPT-G is very similar to HopCPT.
Equation 1 and 3 are still applicable, however, the memory $M$ consists of all available time steps of *all* time series — in
contrast to HopCPT, where only the time series of the predicted basin is used.

Figure 2 illustrates the difference. While in HopCPT only the learned weights are shared between different time steps, in
HopCPT-G both the learned weights as well as the memory vectors are shared between different time series.

This change is also reflected in the training loss. $N$ time steps are drawn without replacement from $K$ randomly selected
time series. The loss for this batch is then calculated as

$$\mathcal{L} = N^{-1} * \|(|\boldsymbol{\epsilon}_{1:N}| - \boldsymbol{A}|\boldsymbol{\epsilon}_{1:N}|)^2\|_1. \tag{4}$$

$$\boldsymbol{A}_{ij} \in \mathbb{R}^{N \times N} = \begin{cases} a_{ij} & \text{for } i \neq j \\ 0 & \text{else.} \end{cases} \tag{5}$$

To ensure that the samples of the own time series (which are likely the most relevant) are in the batch, we choose $N$ and $K$
in a way that all samples of the $K$ series are in the batch.

**PUB Training.** It is crucial for HopCPT-G to learn a time step representation which captures rich information about the error
regime. As shown in multiple works, machine learning models often suffer from shortcut learning (Lapuschkin et al., 2019;
Geirhos et al., 2020). For HopCPT-G a potential shortcut might be to just learn to distinguish the different time series. This
result would especially harm the PUB prediction performance, as it relies on representations which generalize over individual
time series. To facilitate more robust representations that avoid this learning shortcut, we propose a training loss that masks out
all observations of the time series to which the predicted time step belongs. I.e., HopCPT-G can only use the error observations
from the other time series to form its prediction in the training phase. This way, the shortcut of only learning to consider the
"own" time series is impossible. Formally, this changes the association matrix $\boldsymbol{A}$ to

$$\boldsymbol{A}_{ij} \in \mathbb{R}^{N \times N} = \begin{cases} a_{ij} & \text{for } i \neq j \wedge id(i) \neq id(j), \\ 0 & \text{else,} \end{cases} \tag{6}$$

where $id$ maps a time step to an identifier of its corresponding time series.





### 2.2.3 CMAL

CMAL (Klotz et al., 2022) is an LSTM-based mixture density network (Bishop, 1994). The model predicts the parameters of
asymmetric Laplacian distributions. This choice of distribution allows modeling asymmetric uncertainties that are typical for
many hydrological variables. The direct comparison to HopCPT is slightly problematic, as CMAL does not operate on top of
an existing prediction model. However, as one of the best models for uncertainty estimation in streamflow prediction, CMAL
is a good performance yardstick in our evaluation.

### 2.3 Metrics

The evaluation focuses on the validity and efficiency of the prediction intervals of the models. Validity means that, when a
certain coverage, e.g., 90% is specified, also 90% of the test data is actually covered by the prediction interval. This criterion
is measured by the $\Delta$ Cov metric which represents the difference between the specified and the empirical coverage. $\Delta$ Cov is
at best zero, while a notably negative $\Delta$ Cov diminishes the utility of the model. A positive $\Delta$ Cov is less problematic, but a
sign that the model could provide more efficient intervals. Here, efficiency refers to the width of the prediction interval: a more
narrow prediction interval is more efficient than a wider one. We evaluate this property with the PI-Width metric, which simply
corresponds to the average width of the prediction interval over the evaluation period. A smaller PI-Width value is therefore
better. Additionally, we evaluate the Winkler score, which jointly elicits both criteria and thus allows an easy comparison
between different models. The Winkler score is calculated as

$$\mathrm{WS}_\alpha(\boldsymbol{z}_t, M, y_t) = \begin{cases} \mathrm{IW}^\alpha(\boldsymbol{z}_t, M) + \frac{2}{\alpha}(y_t - \widehat{C}^{\alpha,u}(\boldsymbol{z}_t, M)) & \text{if } y_t > \widehat{C}^{\alpha,u}(\boldsymbol{z}_t, M), \\ \mathrm{IW}^\alpha(\boldsymbol{z}_t, M) + \frac{2}{\alpha}(\widehat{C}^{\alpha,l}(\boldsymbol{z}_t, M) - y_t) & \text{if } y_t < \widehat{C}^{\alpha,l}(\boldsymbol{z}_t, M), \\ \mathrm{IW}^\alpha(\boldsymbol{z}_t, M) & \text{else.} \end{cases} \quad (7)$$

The score corresponds to the interval width $\mathrm{IW}_t^\alpha = \widehat{C}^{\alpha,u} - \widehat{C}^{\alpha,l}$ whenever the observed value $y_t$ is within the interval of
$\widehat{C}_t^\alpha(\boldsymbol{Z}_{t+1})$ — else, the score gives penalty that is weighted by the warranted coverage level $\alpha$.[4] The Winkler score as such
calculated for each individual time step $t$, but we report the average over all time steps and basins (as is common in time series
literature).

### 3 Data and Experiments

We evaluate all approaches based on the predictions of an LSTM based rainfall–runoff model. We evaluate all models with three
coverage levels: $\alpha = \{0.05, 0.10, 0.15\}$. Appendix D provides the technical details of our experimental setup; and Appendix E
describes our hyperparameter search.

---

[4] $\widehat{C}^{\alpha,u}$ and $\widehat{C}^{\alpha,l}$ refer to the upper and lower bound of the interval respectively





## 3.1 Data

All experiments are based on the public Catchment Attributes and Meteorology for Large-Sample Studies (CAMELS) data
set (Newman et al., 2015; Addor et al., 2017). CAMELS comprises basis-averaged daily meteorological forcings derived
from three different gridded data products across the United States of America. We used the same 531 basins which were
used in the original benchmark (Newman et al., 2017) and in related follow-up work (Kratzert et al., 2019a; Klotz et al.,
2022). This subset contains basins ranging in size from 4 to $2,000$ km$^2$. The dataset contains daily meteorological forcings
(precipitation, temperature, short-wave radiation, humidity) from three different data sources (NLDAS, Maurer, DayMet),
daily streamflow discharge data from the US Geological Survey, as well as basin-averaged catchment attributes related to soil,
geology, vegetation, and climate. The dataset is split into three parts across the time axis. These parts represent (1) the training
data for the prediction model, (2) the calibration data used by the uncertainty model, and (3) the test data for evaluation. For
CMAL, which does not need any calibration data, both the training and calibration split are used for training.

## 3.2 Experiment I

This experiment assesses how recent measurements affect the performance of HopCPT. We split it into two parts:

**Experiment I-a** compares identical HopCPT models once with "offline" memory and once with "online" memory (see
Section 2.2.1). We examine two different input feature configurations — once using the time series covariates and the model
prediction and once using the model state and model prediction (see Section 2.2.1). In addition to the quantitative comparison,
we qualitatively analyze the individual time series for which the performance gap between "online" and "offline" is the greatest.
This yields insights into potential shifts in the prediction error in such cases.

**Experiment I-b** analyzes intermediate memory update strategies that fall between fully "offline" and fully "online". In real-
world scenarios, gathering the labels with some delay might sometimes be feasible, and at least partially help to mitigate the
impact of distribution shifts. Therefore, we evaluate HopCPT with a memory update frequency of 1 week, 1 month, 3 months,
6 months, 1 year, and 2 years (note that the sampling frequency of the series is one day and the overall test period is 9 years)
and compare the results to the "offline" and fully "online" (i.e., frequency of 1 day).

## 3.3 Experiment II

In Experiment II we investigate how much the data from a given basin contributes towards the uncertainty predictions for said
basin itself. We do so by first comparing HopCPT with HopCPT-G in a gauged setting (Experiment II-a), and then comparing
HopCPT-G with CMAL in an ungauged setting (Experiment II-b). The details for both comparisons are explained in the
following.

**Experiment II-a** compares HopCPT-G to the originally proposed HopCPT in the gauged setting. We focus on the HopCPT
feature configuration which includes the model state and prediction, as we hypothesize that this configuration is less likely to
simply down-weight samples from time series that are different from the predicted one. We additionally evaluate HopCPT-G
with PUB training, i.e., the adapted training loss (see Equation 6). Although we do not evaluate on ungauged basins in this





setting, PUB training can increase the tendency of HopCPT to incorporate data from other basins — and that could potentially
lead to more robust representations.

**Experiment II-b** investigates if uncertainty estimates are also possible without any local information from the predicted
basin. To examine this scenario we loosely follow the PUB setting from Kratzert et al. (2019a): The set of time series is split
into 11 mutually exclusive subsets of equal size. The principle is similar to k-fold cross validation (but not on a per-sample
basis): We define that for the k-th fold, the gauged basins are the union of all but the k-th subset, while the k-th subset represents
the ungauged basins. We reserve one of the folds for hyperparameter tuning and exclude it from the evaluation. For the other
10 folds, we individually train the prediction and uncertainty model on the "gauged" basins (9 out of 10 subsets) and evaluate
them on the ungauged basins from the remaining subset. Note that within each subset, each time series is (as in the standard
case) split into training, calibration, and test data. CMAL's training data encompasses both the training and calibration split
to ensure that in total each model has the same amount of data available. We evaluate only the test period of the ungauged
basins (this avoids information leakage from the train and calibration period of the gauged basins). We evaluate two variants
of HopCPT: (1) HopCPT-G with normal training and (2) HopCPT-G with PUB training. For both variants, we use the model
state and prediction as input features.

## 3.4   Experiment III

Experiment III examines the uncertainty estimation for peak flows specifically. Peak flow uncertainties are especially hard to
capture. Firstly, because prediction models tend to make larger errors (i.e., high aleatoric uncertainty); and secondly because
the occurrence of peak flow events is limited (i.e., high epistemic uncertainty). To measure the respective performance we
calculate the metrics only using time steps where the streamflow observations are in the top $x$ % of the corresponding basin.
Specifically, we evaluate for $x \in \{2, 5, 10, 20, 30, 50, 100\}$ (and 100 comprises all data and hence corresponds to the "standard"
evaluation) within the gauged basin setting of Experiment II. For the sake of completeness, Appendix G3 also presents the
peak flow evaluation for the other experiments.

## 4   Results and Discussion

### 4.1   Experiment I

**Experiment I-a** compares the offline and online mode of HopCPT. Table 1 shows that the main performance advantage of the
online setting is its better coverage. This holds especially true for the HopCPT variant which uses the model states as features.
The width of the prediction interval stays almost constant between offline and online approaches. This suggests that the change
in Winkler score is a direct effect of the lower/higher coverage. We argue that the slight loss in coverage for the offline setting
is due to a distribution shift in some basins (Fig. 3): While the coverage distribution of the basins with highest coverage is very
similar between the offline and online cases, the biggest change happens in the $10-20\%$ of basins with the lowest coverage.





**Table 1.** HopCPT performance of the two evaluated input combinations in offline and online mode for the miscoverage levels $\alpha = \{0.05, 0.10, 0.15\}$. The values represent the average over all basins. Bold numbers correspond to the best result for the respective metric in the experiment (PI-Width and Winkler score) — significance tested with a Mann–Whitney $U$ test at $p < 0.005$. For PI-Width and Winkler score lower values are better — for $\Delta$ Cov non-negative values close to 0 are best. The values in parenthesis represent the standard deviation over the different seeds.

| | $\alpha$ | | 0.05 | | | 0.10 | | | 0.15 | |
|---|---|---|---|---|---|---|---|---|---|---|
| | | $\Delta$ Cov | PI-Width | Winkler | $\Delta$ Cov | PI-Width | Winkler | $\Delta$ Cov | PI-Width | Winkler |
| X / YHat | offline | −.005 (0.002) | **2.88** (0.09) | 1.21 (0.02) | −.011 (0.003) | **2.15** (0.06) | 0.94 (0.01) | −.017 (0.003) | **1.75** (0.05) | 0.80 (0.01) |
| | online | .005 (0.002) | **2.87** (0.09) | **1.16** (0.02) | .004 (0.002) | **2.14** (0.06) | **0.91** (0.01) | .001 (0.002) | **1.74** (0.05) | **0.78** (0.01) |
| Model States / YHat | offline | −.017 (0.003) | **1.92** (0.06) | 1.02 (0.02) | −.029 (0.004) | **1.44** (0.05) | 0.78 (0.01) | −.039 (0.005) | **1.17** (0.04) | 0.66 (0.01) |
| | online | .005 (0.002) | **1.96** (0.06) | **0.90** (0.01) | .002 (0.003) | **1.46** (0.04) | **0.71** (0.01) | −.004 (0.004) | **1.19** (0.04) | **0.61** (0.00) |

A particularly striking examples of this shift as shown in Figure 4: In the prediction year 2000 (approx. 1 year after the memory end), the offline setting provides reasonable intervals. However, in the year 2004, the real streamflow seems to be shifted upwards for low flows and downwards for some higher flows. Since the output of the prediction model remains roughly constant, we suggest that the phenomenon at hand is a shift in the runoff that not visible in the input patterns. The online model can accommodate for this shift since the new information (in the form of the shifted observations) are incorporated into the

memory. The offline model simply has no mechanism to account for that and construct invalid intervals. Figures G1 and G2 in the appendix show additional examples of this error shift behavior.

**Experiment I-b** investigates the effect of HopCPT with infrequent memory updates, as intermediate settings between the edge cases of fully online and offline settings. Table 2 presents the results. The slight coverage loss increases gradually with a higher update delay — as illustrated in Figure 5. However, an update frequency of 1 year already halves the coverage loss

compared to the offline setting. The changes in the PI-Width are less clear and non-monotonic, however, the variation is anyway negligible.

---

**Answer to RQI: Does up to date information improve the general uncertainty estimation?**

Our results indeed suggest that a continuous incorporation of new data improves uncertainty predictions. For the center of the predictive distribution (associated with large $\alpha$ values; Tab. 1) our results are less pronounced than for the tails (associated with small $\alpha$ values; Tab. 1). Further, continuously updating the uncertainty estimates as new data comes along is very useful, given that environmental processes are associated with all kinds of distribution shifts. In our experiments, a memory update mechanism was advantageous even in cases where real-world constraints only allow for very infrequent updates (Tab. 2).

---





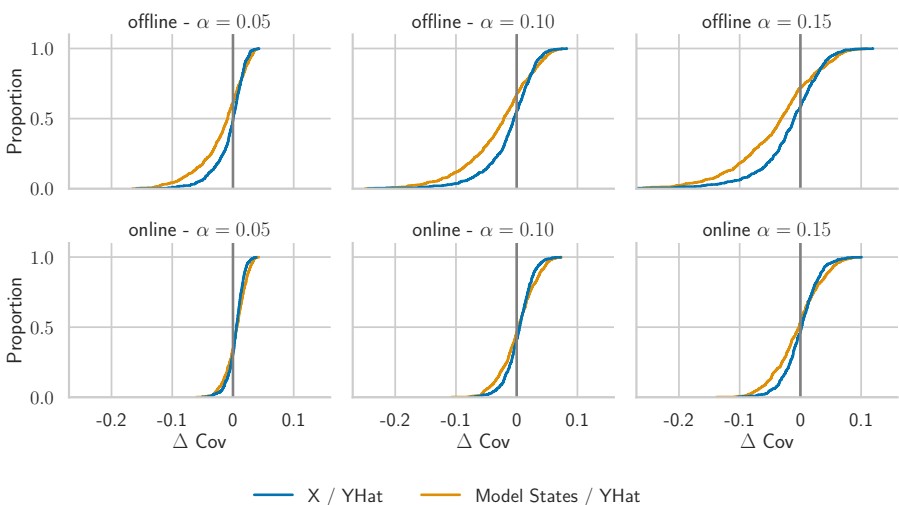

**Figure 3.** CDF of $\Delta$ Cov over individual basins for the models evaluated in Experiment I-a. The bottom row shows the results for the "online" mode, i.e., the memory gets updated after each prediction, and the top row for the "offline" mode, i.e., the memory does not get updated during the test period. For $\Delta$ Cov non-negative values close to 0 are best.

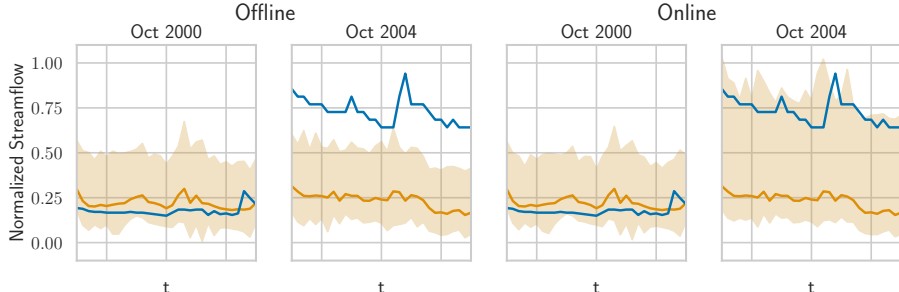

**Figure 4.** The real streamflow (blue), the prediction (gold), and HopCPT's prediction interval (light gold) for a basin (id: 12447390) in October 2000 and October 2004. The two plots one the left show HopCPT in the offline mode — the right two plots in the online mode. The online mode allows HopCPT to account for the distribution shift in October 2004.





**Table 2.** Performance of HopCPT with different memory update behaviour for the miscoverage levels $\alpha = \{0.05, 0.10, 0.15\}$. The "Delay" column indicates the update frequency of the memory. The values represent the average over all basins. Bold numbers correspond to the best result for the respective metric in the experiment (PI-Width and Winkler score). For PI-Width and Winkler score lower values are better — for $\Delta$ Cov non-negative values close to 0 are best. The values in parenthesis represent the standard deviation over the different seeds.

| Delay | 0.05 | | | 0.10 | | | 0.15 | | |
|---|---|---|---|---|---|---|---|---|---|
| | $\Delta$ Cov | PI-Width | Winkler | $\Delta$ Cov | PI-Width | Winkler | $\Delta$ Cov | PI-Width | Winkler |
| Online | 0.005 | **1.96** | **0.900** | 0.002 | **1.46** | **0.710** | −0.004 | **1.19** | **0.610** |
| | (0.002) | (0.06) | (0.010) | (0.003) | (0.04) | (0.010) | (0.004) | (0.04) | (0.000) |
| 1 Week | 0.002 | **1.98** | 0.922 | −0.002 | **1.47** | **0.719** | −0.009 | **1.20** | 0.617 |
| | (0.002) | (0.05) | (0.009) | (0.002) | (0.04) | (0.006) | (0.003) | (0.03) | (0.004) |
| 1 Month | −0.002 | **1.97** | 0.943 | −0.007 | **1.47** | 0.731 | −0.014 | **1.20** | 0.625 |
| | (0.002) | (0.05) | (0.010) | (0.003) | (0.04) | (0.006) | (0.003) | (0.03) | (0.004) |
| 3 Months | −0.005 | **1.97** | 0.956 | −0.011 | **1.47** | 0.739 | −0.019 | **1.20** | 0.631 |
| | (0.002) | (0.05) | (0.010) | (0.003) | (0.04) | (0.006) | (0.003) | (0.03) | (0.004) |
| 6 Months | −0.006 | **1.97** | 0.961 | −0.013 | **1.47** | 0.742 | −0.021 | **1.20** | 0.633 |
| | (0.002) | (0.05) | (0.010) | (0.003) | (0.04) | (0.006) | (0.003) | (0.03) | (0.004) |
| 1 Year | −0.007 | **1.97** | 0.964 | −0.014 | **1.47** | 0.743 | −0.023 | **1.20** | 0.634 |
| | (0.002) | (0.05) | (0.010) | (0.003) | (0.04) | (0.006) | (0.003) | (0.03) | (0.004) |
| 2 Years | −0.008 | **1.97** | 0.969 | −0.016 | **1.47** | 0.747 | −0.025 | **1.20** | 0.637 |
| | (0.002) | (0.05) | (0.011) | (0.003) | (0.04) | (0.007) | (0.003) | (0.03) | (0.005) |
| Offline | −0.017 | **1.92** | 1.020 | −0.029 | **1.44** | 0.780 | −0.039 | **1.17** | 0.660 |
| | (0.003) | (0.06) | (0.020) | (0.004) | (0.05) | (0.010) | (0.005) | (0.04) | (0.010) |

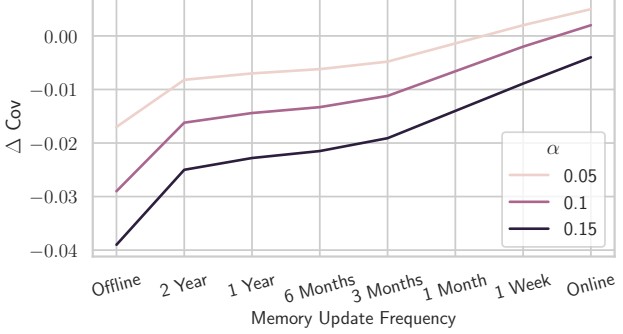

**Figure 5.** $\Delta$ Cov of HopCPT for different memory update frequency settings. Increasing update frequency monotonically improves the coverage for each evaluated coverage level. For $\Delta$ Cov non-negative values close to 0 are best.





## 4.2 Experiment II

**Experiment II-a** probes the effect of global data in the "gauged" basin setting. Table 3 shows the result of HopCPT-G (with and without PUB training) and compares it to the non-global HopCPT. Suprisingly, the shared memory of HopCPT-G (no PUB training) does not improve the results. However, it also does not do any notable harm. We hypothesize that the calibration data of each basin is already so comprehensive that all relevant error regimes for the respective basin are covered with sufficient resolution. The slight degradation in performance could either result from (i) the additional challenge of learning to disregard

all non-relevant basins, or (ii) the missing temporal encoding in HopCPT-G. The PUB training procedure, on the other hand, notably improves the coverage. However, this improvement comes at the cost of efficiency. Hence, the resulting Winkler Score is similar to the other approaches.

Figure 6 shows the coverage distribution over the individual basins. The distributions of HopCPT and HopCPT-G are very similar. This indicates that HopCPT-G cannot profit from the cross-time-series information, but successfully learns to exclude

irrelevant basins in order to arrive at the "right" error distribution.

**Experiment II-b** investigates the PUB setting. Table 4 shows the result of the evaluation. CMAL, which has very good coverage in a gauged basin setting (see Appendix A), exhibits a noticeable under-coverage. HopCPT-G, on the other hand, only slightly loses coverage compared to the non-PUB setting. HopCPT-G with PUB training reduces the coverage loss notably. Its efficiency is comparable to CMAL and HopCPT-G. CMAL achieves the best Winkler score despite its high under-coverage.

This suggests that the uncovered samples are relatively close to the border of the prediction interval. Figure 7 shows the distribution of $\Delta$ Cov over the individual time series. Since the PUB setting does still yield acceptable performances but does not allow to use information from a given basin, one can conclude that the global setting is able to transfer uncertainty information about the uncertainty from other basins to the unseen basins. Quantitative metrics for the individual folds can be found in the appendix in Table G1.


> **Answer to RQII: Is the data from a given basin required to get good uncertainty estimates?**
>
> Our result suggests that local information is beneficial for efficient uncertainty intervals. Given that enough local information is available and the average quality of the estimate is the focus (in contrast to RQIII), it is also sufficient to provide reasonable uncertainty estimates (Experiment II-a). However, local information is not strictly necessary to produce sensible uncertainty estimations, and information transfer via global information is possible (Experiment II-b).





**Table 3.** Performance of the different HopCPT variants with for the miscoverage levels $\alpha = \{0.05, 0.10, 0.15\}$. The values represent the average over all basins. Bold numbers correspond to the best result for the respective metric in the experiment (PI-Width and Winkler score) — significance tested with a Mann–Whitney $U$ test at $p < 0.005$. For PI-Width and Winkler score lower values are better — for $\Delta$ Cov non-negative values close to 0 are best. The values in parenthesis represent the standard deviation over the different seeds.

| $\alpha$ | | 0.05 | | | 0.10 | | | 0.15 | |
|---|---|---|---|---|---|---|---|---|---|
| | $\Delta$ Cov | PI-Width | Winkler | $\Delta$ Cov | PI-Width | Winkler | $\Delta$ Cov | PI-Width | Winkler |
| HopCPT | $-.017$ | **1.92** | **1.02** | $-.029$ | **1.44** | **0.78** | $-.039$ | **1.17** | **0.66** |
| | (0.003) | (0.06) | (0.02) | (0.004) | (0.05) | (0.01) | (0.005) | (0.04) | (0.01) |
| HopCPT-G | $-.023$ | 2.10 | 1.06 | $-.031$ | 1.61 | 0.81 | $-.036$ | 1.35 | 0.68 |
| | (0.003) | (0.07) | (0.01) | (0.004) | (0.05) | (0.00) | (0.004) | (0.05) | (0.00) |
| HopCPT-G | .018 | 3.16 | **1.04** | .021 | 2.36 | **0.81** | .019 | 1.93 | 0.69 |
| (PUB Train) | (0.006) | (0.18) | (0.07) | (0.008) | (0.13) | (0.04) | (0.011) | (0.11) | (0.02) |

**Table 4.** Performance of different models on the PUB experiment for the miscoverage levels $\alpha = \{0.05, 0.10, 0.15\}$. The values represent the average over all test basins of all splits. Bold numbers correspond to the best result for the respective metric in the experiment (PI-Width and Winkler score) — significance tested with a Mann–Whitney $U$ test at $p < 0.005$. For PI-Width and Winkler score lower values are better — for $\Delta$ Cov non-negative values close to 0 are best. The values in parenthesis represent the standard deviation over the different seeds.

| $\alpha$ | | 0.05 | | | 0.10 | | | 0.15 | |
|---|---|---|---|---|---|---|---|---|---|
| | $\Delta$ Cov | PI-Width | Winkler | $\Delta$ Cov | PI-Width | Winkler | $\Delta$ Cov | PI-Width | Winkler |
| HopCPT-G | $-.052$ | **2.15** | 1.78 | $-.070$ | **1.57** | 1.29 | $-.080$ | **1.27** | 1.05 |
| | (0.001) | (0.01) | (0.02) | (0.001) | (0.01) | (0.01) | (0.002) | (0.01) | (0.01) |
| HopCPT-G | $-.007$ | 3.00 | 1.34 | $-.017$ | 2.27 | 1.04 | $-.026$ | 1.87 | 0.89 |
| (PUB Train) | (0.003) | (0.08) | (0.04) | (0.005) | (0.07) | (0.03) | (0.006) | (0.06) | (0.02) |
| CMAL | $-.119$ | 2.44 | **1.18** | $-.155$ | 1.91 | **0.93** | $-.172$ | 1.61 | **0.80** |
| | (0.008) | (0.08) | (0.03) | (0.009) | (0.06) | (0.02) | (0.009) | (0.05) | (0.02) |





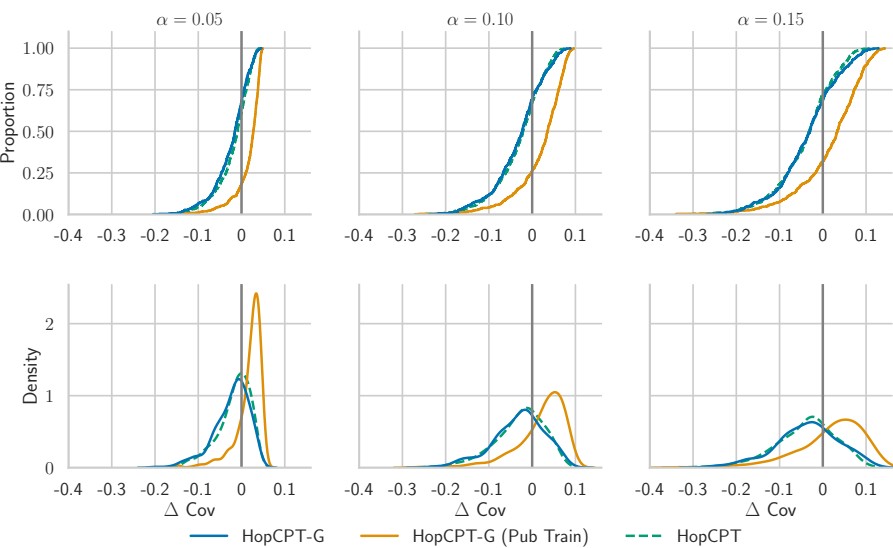

**Figure 6.** CDF and PDF of $\Delta$ Cov over individual basins for models evaluated in Experiment II-a. For $\Delta$ Cov non-negative values close to 0 are best.

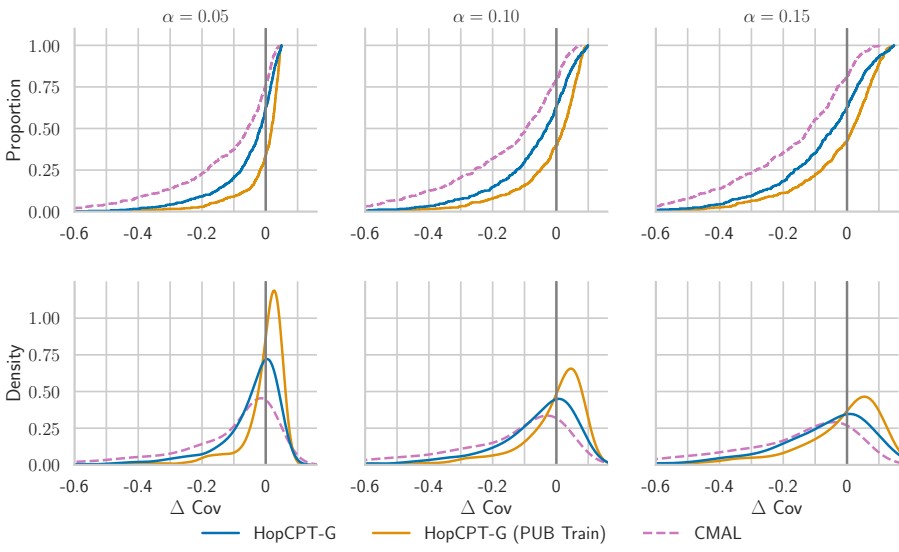

**Figure 7.** CDF and PDF of $\Delta$ Cov over all individual test basins of all PUB folds for models evaluated in Experiment II-b. For $\Delta$ Cov non-negative values close to 0 are best.



### 4.3 Experiment III

In contrast to our results from Experiment II, the peak flow performance exhibits a notable difference between the HopCPT variants (Fig. 8). HopCPT-G provides better coverage than HopCPT, PUB training boosts this further. The coverage difference
between the HopCPT variants also increases with increasing $\alpha$. CMAL, which is also a "fully global" model, achieves the best Winkler scores in the peak flow setting. This indicates that the information from other basins is beneficial for peak flow regimes — even when plenty of past information for the basin is available. We argue that this is because the data is more scarce in these regimes and certain situations might not be available in the observed past of the individual basin. Thus, in this situation it becomes useful to leverage information from the other basins in order to obtain good uncertainty estimates (we note that
this is in contrast to the results from Experiment II, which showed that for the average runoff predictions no improvement is obtained by considering global information).

> **Answer to RQIII: Is data from a single basin enough to get good uncertainty predictions for peak flows?**
> Our results indicate that using only data from a single basin — that is, only considering the local information — leads to worse uncertainty estimates for peak flow settings. The more restricted the "peak flow categorization" is chosen, the more pronounced this effect gets (Figure 8). Using global information — i.e., information from other basins — improves the uncertainty estimates. In particular, this holds in terms of reliability of the estimate (i.e., coverage).

### 5  Conclusions and Outlook

This contribution investigates how the temporal (recency) and spatial (local vs. global) dimensions of information impact the quality of uncertainty estimates. To conduct our study we apply the Conformal Prediction (CP) framework in the form of the Hopfield Conformal Prediction for Timeseries (HopCPT) approach, which extends the CP framework for time series predictions. In short, we find that (a) the inclusion of the most recent information has high benefits for the general uncertainty predictions; and (b) that global information is not important for general uncertainty predictions, but pivotal to provide good
bounds for peak flows.

Regarding (a), our analysis suggests that incorporating recent information helps to improve the uncertainty estimates. This is even true if the information can only be provided after longer periods (e.g., when for technical reasons only yearly updates are possible). We could qualitatively link this phenomenon to distribution shifts that appear over time. We conclude that continuous monitoring and incorporation of the newly obtained data in the prediction process are vital components of prediction systems
that strive to provide reliable and efficient uncertainty estimations.

Regarding (b), our results indicate that local information is sufficient to provide good uncertainty estimates on average (assuming that a reasonable history over multiple years is provided). However, for peak flows it is not. We argue that this is because the estimation problem is particularly hard since high-flow situations are diverse, have a high measurement variance, and happen very infrequently. Global information is able to improve the estimates. These observations are in accordance with
the results from Frame et al. (2022) and Bertola et al. (2023) — which indicate that the signals from different basins can be



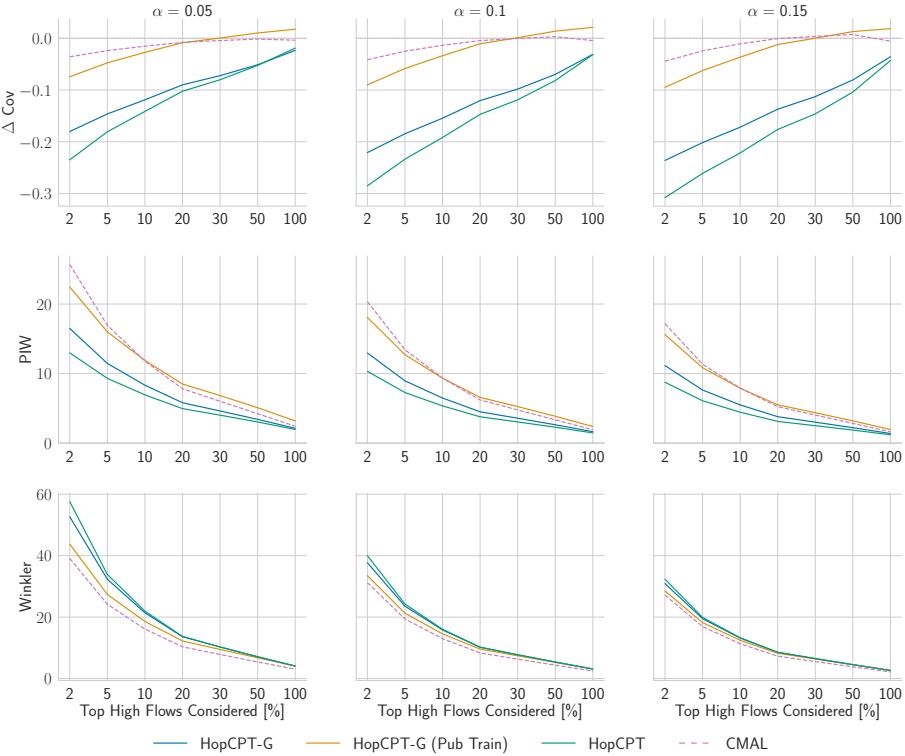

**Figure 8.** Evaluation metrics for Experiment III. Only peak flow time steps — defined via varying share of the overall steps (x-axis of the plots) — are considered for the respective metric. For PI-Width and Winkler score lower values are better — for $\Delta$ Cov non-negative values close to 0 are best.

leveraged to make better predictions for peak flows. This can be taken to the extreme for predictions in ungauged basins. Indeed, our experiments in this regard suggest that reasonable uncertainty estimates can be also provided without any local information. However, whenever global information is used for uncertainty estimation it is especially important that the respective model can selectively consider the relevant information — i.e. can decide which parts of the overall information should be used for the "current" estimation. HopCPT-G and CMAL are both able to do that.

In future work, experiments with more scarce local data could reveal further advantages of approaches that incorporate global information. More broadly, CP offers a fresh perspective to uncertainty-aware streamflow prediction. Appendix B shows a first exploration of how the principles from CP can improve existing hydrological uncertainty estimations with minimal interventions. In the future, we want to explore how ideas from CP are able to refine other (perhaps less formal) approaches that are currently used in practice.



*Code and data availability.* The code repository is available at https://doi.org/10.5281/zenodo.10674231. The trained base model (LSTM) and the utilized model states as well as the global HopCPT models are available at https://doi.org/10.5281/zenodo.10653863. The trained CMAL models for the non-PUB experiments is available at https://doi.org/10.5281/zenodo.10654345, the CMAL models for the PUB experiments are available at https://doi.org/10.5281/zenodo.10654399. The data for CAMELS can be accessed for free on the NCAR's official website (https://gdex.ucar.edu/dataset/camels/file.html). The expanded Maurer forcings, which include data on daily minimum and maximum temperatures, are available for download at https://doi.org/10.4211/hs.17c896843cf940339c3c3496d0c1c077 (Kratzert, 2019).

.





## Appendix: Overview

The appendix is structured as follows: Appendix A compares the overall performance of different uncertainty methods, in-
cluding HopCPT and CMAL. Appendix B investigates the impact of applying the principles of CP to the existing uncertainty
estimation method Bluecat. Appendix C investigates how different input features influence the uncertainty estimation perfor-
mance — i.e. provides an orthogonal dimension to the spatial and temporal consideration of the main paper. Appendix D and
Appendix E provide details about the experimental setup and the hyperparameter tuning respectively. Appendix F presents the
point prediction metrics of the experiments. Finally, Appendix G provides additional results to the experiments in the main
paper.

## Appendix A: Model intercomparison

This section compares the performance of the different (non-global methods[5]). For HopCPT, we evaluate the model offline
(i.e., without updating the memory as outlined in Section 2.2.1). Two variants for the input features are examined: (1) The time
series features and the model prediction, similar to the original work but without the lagged target; (2) the model states and
prediction as proposed in Section 2.2.1. For Bluecat, we selected the best model considering both the original Blucat and the
adapted version suggested in Sec. B. We follow Auer et al. (2023) and also evaluate against kNN as a naive similarity-based
baseline (see Appendix A1). While Auer et al. (2023) show that kNN on the time series features is not sufficient, we analyze the
performance of kNN with (1) the model states and prediction (corresponds to HopCPT) and (2) with only the model prediction
(corresponds to Bluecat) as input. Additionally, we evaluate CMAL as the current state-of-the-art approach.

### A1   kNN as Navive Similarity Approach

HopCPT uses the similarity representation that is learned by the MHN network. A more naive way to consider such a similarity
would be to simply use a kNN model based on the model inputs. Following the original work, we use such a kNN model as a
baseline in the experiments.

### A2   Model intercomparison - Results

Table A1 shows the evaluation of the different models. CMAL, closely followed by HopCPT, provides the best coverage at
all coverage levels. While HopCPT with the model states and input reaches the most efficient — i.e., smallest — prediction
intervals, CMAL performs best in terms of the Winkler score. This is slightly surprising, given that the Winkler score encom-
passes the coverage and PI-Width. However, the fact that Winkler score additionally considers the distance of the uncovered
test samples leads us to the hypothesis that CMAL, as it is optimized to maximize the likelihood of all samples, results in
smaller distances in that regard. In contrast, HopCPT and Bluecat do not consider the uncovered samples in optimization and
calibration. The comparison between the HopCPT evaluations with different input features shows that adding the model states

---

[5]CMAL can be considered a global model





**Table A1.** Performance of the evaluated models for the miscoverage levels $\alpha = \{0.05, 0.10, 0.15\}$. The values represent the average over all basins. Bold numbers correspond to the best result for the respective metric in the experiment (PI-Width and Winkler score), given that $\Delta$ Cov $\leq -\alpha$, i.e., the specific algorithm reached reasonable coverage (the result is grayed otherwise) — significance tested with a Mann–Whitney $U$ test at $p < 0.005$. The values in parenthesis represent the standard deviation over the different seeds (results without these are from deterministic models).

| $\alpha$ | 0.05 | | | 0.10 | | | 0.15 | | |
|---|---|---|---|---|---|---|---|---|---|
| | $\Delta$ Cov | PI-Width | Winkler | $\Delta$ Cov | PI-Width | Winkler | $\Delta$ Cov | PI-Width | Winkler |
| HopCPT | −.005 | 2.88 | 1.21 | −.011 | 2.15 | 0.94 | −.017 | 1.75 | 0.80 |
| | (0.002) | (0.09) | (0.02) | (0.003) | (0.06) | (0.01) | (0.003) | (0.05) | (0.01) |
| HopCPT | −.017 | **1.92** | 1.02 | −.029 | **1.44** | 0.78 | −.039 | **1.17** | 0.66 |
| (model states) | (0.003) | (0.06) | (0.02) | (0.004) | (0.05) | (0.01) | (0.005) | (0.04) | (0.01) |
| Bluecat′ | −.050 | 2.72 | 1.19 | −.057 | 2.00 | 0.90 | −.061 | 1.63 | 0.76 |
| (M=100) | | | | | | | | | |
| kNN | −.176 | 1.95 | 1.84 | −.202 | 1.47 | 1.26 | −.215 | 1.19 | 1.01 |
| (yHat) | | | | | | | | | |
| kNN | −.193 | 1.37 | 1.74 | −.219 | 1.04 | 1.15 | −.231 | 0.85 | 0.90 |
| (model states) | | | | | | | | | |
| CMAL | −.004 | 2.40 | **0.78** | −.004 | 1.89 | **0.63** | −.006 | 1.60 | **0.55** |
| | (0.004) | (0.06) | (0.01) | (0.008) | (0.05) | (0.01) | (0.011) | (0.04) | (0.01) |

as inputs enhances the efficiency of HopCPT while keeping an approximate coverage, which results in an overall lower Winkler score. A more detailed analysis of the effect of different input modalities is given in Appendix C. The best variant of Bluecat falls behind HopCPT and CMAL in all three metrics. The kNN model, as a naive similarity measure baseline, results in high

420 under-coverage also when considering the model states. This reinforces the motivation for a learned similarity measure, as already mentioned in the original HopCPT paper.

**Individual basin analysis** Figure A1 gives more detailed insights into how the metrics are distributed around the individual basins. CMAL, which archives the best overall coverage, does not provide better coverage for all basins but the high mean is driven by the higher over-coverage of the upper half of basins. This aligns with the analysis in (Klotz et al., 2022). Overall,

425 the coverage distribution of HopCPT is more centered around zero and better bounds the lowest coverage. As the cumulative distribution plot in Figure A2 shows, for the 50 (approx 10%) basins with worst coverage, HopCPT without model states performs best, and the worst coverage is best for HopCPT (no matter if with or without model states). The distribution of the PI-Width (Figure A3) is similar for all models, however, the efficiency advantage of HopCPT is most pronounced for the basins with larger prediction intervals.





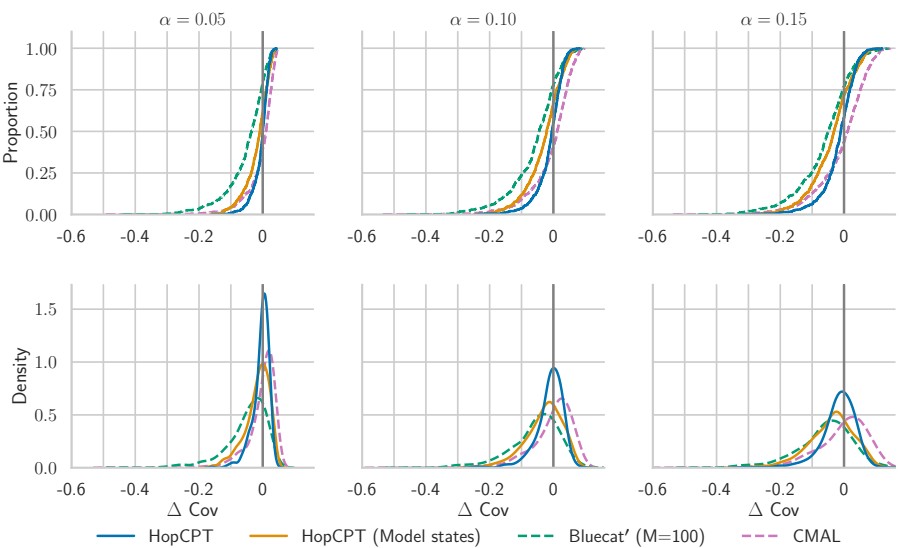

**Figure A1.** CDF and PDF of $\Delta$ Cov over individual basins for models with approximate average coverage ($\Delta$ Cov $\leq -\alpha$) evaluated in Appendix A.

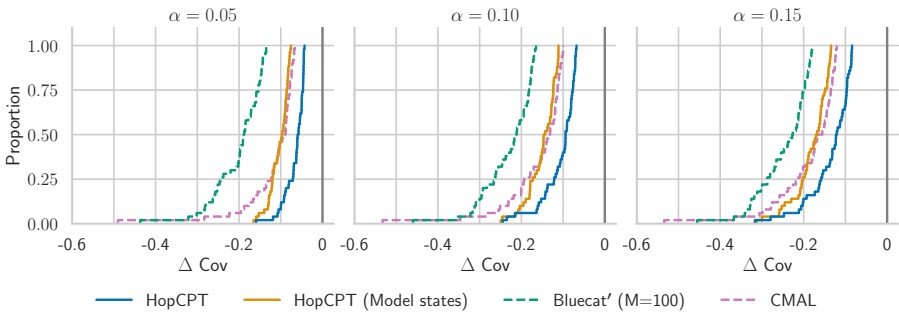

**Figure A2.** CDF of $\Delta$ Cov over the 50 basins with the highest miscoverage for models with approximate average coverage ($\Delta$ Cov $\leq -\alpha$) evaluated in Appendix A.





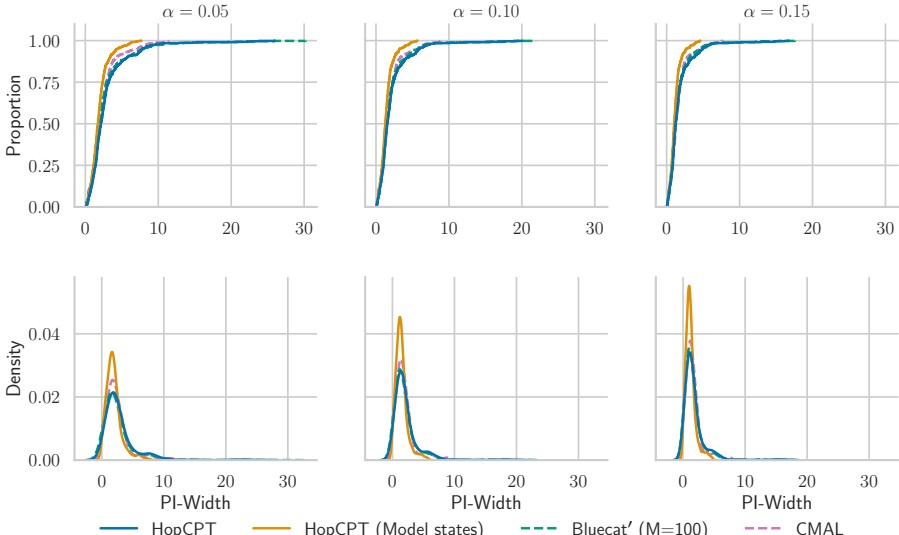

**Figure A3.** CDF and PDF of PI-Width over individual basins for models with approximate average coverage ($\Delta$ Cov $\leq -\alpha$) evaluated in Appendix A.

## Appendix B: Bluecat

Similar to HopCPT, the hydrological uncertainty estimation approach Bluecat (Koutsoyiannis and Montanari, 2022) aims to provide a prediction interval given a point prediction model and a set of data points (calibration data). Bluecat first evaluates the calibration data. Then, for a new prediction point, the calibration point with the most similar model prediction and all points within a certain distance to this point are considered. This can be seen as a special form of kNN where (1) the distance metric between two points is the distance of the model predictions and (2) not k points but all points within a certain distance threshold are considered (Rozos et al., 2022). Given this set of selected calibration points, similar to the CP framework, the specified quantiles are calculated and provide the bounds of the confidence band.

**The CP perspective on Bluecat (Bluecat′)**

Viewing Bluecat through a CP lens suggests a relatively low effort improvement in the form of the introduction of a calibration set for the interval prediction. Bluecat only uses the training data of the prediction model to determine the prediction interval, i.e., prediction training data = calibration data. However, the CP literature suggests that an independent calibration set can give a more unbiased estimate, since prediction models can easily overfit training data (and, as a result, the prediction intervals might be overly optimistic and do not fulfill the specified coverage criterion). We refer to this adaptation as Bluecat′ throughout the paper.

We evaluate Bluecat once as originally proposed (Bluecat) and once with the adaption (Bluecat′). Bluecat requires setting the hyperparameter $M$, which controls how many close data points are considered. In the original publication, there is no clear guidance on how to select the parameter for new datasets, which is why we evaluate two variations: (1) tuning the





**Table B1.** Performance of the evaluated Bluecat variants for the miscoverage levels $\alpha = \{0.05, 0.10, 0.15\}$. The values represent the average over all basins. Bold numbers correspond to the best result for the respective metric in the experiment (PI-Width and Winkler score). As Bluecat is deterministic no standard deviation is given.

| $\alpha$ | 0.05 | | | 0.10 | | | 0.15 | | |
|---|---|---|---|---|---|---|---|---|---|
| | $\Delta$ Cov | PI-Width | Winkler | $\Delta$ Cov | PI-Width | Winkler | $\Delta$ Cov | PI-Width | Winkler |
| Bluecat (M=100) | $-.092$ | **2.09** | 1.51 | $-.112$ | **1.55** | 1.06 | $-.122$ | **1.26** | 0.86 |
| Bluecat (M=225) | $-.071$ | 2.22 | 1.55 | $-.091$ | 1.60 | 1.10 | $-.102$ | 1.28 | 0.89 |
| Bluecat′ (M=100) | $-.050$ | 2.72 | **1.19** | $-.057$ | 2.00 | **0.90** | $-.061$ | 1.63 | **0.76** |
| Bluecat′ (M=225) | $-.028$ | 2.88 | 1.24 | $-.035$ | 2.06 | 0.93 | $-.039$ | 1.64 | 0.78 |

hyperparameters with the same model selection criteria as in HopCPT (resulting in $M = 225$), and (2) using the hyperparameter value from the original publication ($M = 100$).

## B1  Bluecat - Results

Table B1 shows the result of the comparison. The adaptation Bluecat′ achieves considerably improved coverage. The lower Winkler scores further indicate the overall better performance of Bluecat′. These results support the hypothesis that Bluecat intervals are overly optimistic since they are based on the — most likely overfitted — training data predictions of the prediction model. Further, the hyperparameter-tuned versions ($M = 225$) of both Bluecat variants enhance the coverage of the models at the cost of lower interval efficiency. This makes sense, as the model selection favors models with better coverage as long as no model fully achieves the specified coverage.

## Appendix C:  Input information

This section assesses how different HopCPT inputs affect its performance. The experiment compares all combinations of the three potential feature components: time series covariates ($X$), model prediction (YHat), and model state. We use an LSTM prediction model. Hence, the model state refers to the two internal state vectors of the LSTM, which are updated in each new step and are — together with the current input — the basis for the output of the model. Specifically, we consider the state vectors right after the prediction, since these states already consider the current input features. We tuned hyperparameters individually for each feature set combination. Additionally, we analyze results where we include the lagged target (Y) as a feature, as in the original HopCPT paper.





**Table C1.** HopCPT performance of the evaluated input combinations for the miscoverage levels $\alpha = \{0.05, 0.10, 0.15\}$. The values represent the average over all basins. Bold numbers correspond to the best result for the respective metric in the experiment (PI-Width and Winkler score) — significance tested with a Mann–Whitney $U$ test at $p < 0.005$. The values in parenthesis represent the standard deviation over the different seeds.

| $\alpha$ | 0.05 | | | 0.10 | | | 0.15 | | |
|---|---|---|---|---|---|---|---|---|---|
| | $\Delta$ Cov | PI-Width | Winkler | $\Delta$ Cov | PI-Width | Winkler | $\Delta$ Cov | PI-Width | Winkler |
| X | $-.003$ | 3.50 | 1.61 | $-.005$ | 2.35 | 1.17 | $-.007$ | 1.80 | 0.96 |
| | (0.002) | (0.15) | (0.02) | (0.003) | (0.10) | (0.01) | (0.005) | (0.08) | (0.01) |
| X / YHat | $-.005$ | 2.88 | 1.21 | $-.011$ | 2.15 | 0.94 | $-.017$ | 1.75 | 0.80 |
| | (0.002) | (0.09) | (0.02) | (0.003) | (0.06) | (0.01) | (0.003) | (0.05) | (0.01) |
| Model States | $-.017$ | **1.93** | **1.01** | $-.029$ | **1.45** | **0.77** | $-.041$ | **1.18** | **0.66** |
| | (0.004) | (0.06) | (0.01) | (0.005) | (0.04) | (0.01) | (0.006) | (0.03) | (0.00) |
| Model States / YHat | $-.017$ | **1.92** | **1.02** | $-.029$ | **1.44** | **0.78** | $-.039$ | **1.17** | **0.66** |
| | (0.003) | (0.06) | (0.02) | (0.004) | (0.05) | (0.01) | (0.005) | (0.04) | (0.01) |
| X / Model States | $-.019$ | **1.90** | **1.02** | $-.032$ | **1.43** | **0.78** | $-.043$ | **1.17** | **0.66** |
| | (0.003) | (0.04) | (0.02) | (0.005) | (0.03) | (0.01) | (0.006) | (0.03) | (0.01) |
| X / Model States / YHat | $-.020$ | **1.90** | **1.01** | $-.032$ | **1.43** | **0.77** | $-.043$ | **1.18** | **0.65** |
| | (0.004) | (0.05) | (0.01) | (0.005) | (0.04) | (0.01) | (0.005) | (0.03) | (0.01) |

Table C1 shows the performance of HopCPT with the different input feature configurations. Interestingly, only using the time series covariates results in the best coverage. However, the prediction intervals in these settings are rather large, which is also reflected in notably higher Winkler scores. Therefore, we argue that using the model states as input features is indeed beneficial. This argument is further strengthened by the results in Experiment I, where we see that the coverage loss is likely due to distribution shifts. Accounting for this shift by updating the memory leads to almost perfect coverage for the model

state setting and unnecessary over-coverage in the covariate-only setting. Both adding the model prediction and the model state provide additional information. The model state does not only include information about the current covariates, but also includes history information due to the recursive nature of the LSTM. The model prediction, on the other hand, can be seen as a projection of the model state where some relevant information could be lost. Extending the model state with additional inputs hardly changes the performance of HopCPT, which supports the hypothesis that the model state already includes the

vast majority of the required information.

     Table C2, shows the results when the lagged target variable is included in the feature set, as done in the original work Auer et al. (2023). Note that this is typically not feasible in streamflow prediction. However, to how this affects the performance we also evaluated these input combinations as it is possible in the lab setting.





**Table C2.** HoCPT performance of the input combinations which include the lagged target $y$ for the miscoverage levels $\alpha = \{0.05, 0.10, 0.15\}$. The values represent the average over all basins. Bold numbers correspond to the best result for the respective metric in the experiment (PI-Width and Winkler score). The values in parenthesis represent the standard deviation over the different seeds.

| $\alpha$ | 0.05 | | | 0.10 | | | 0.15 | | |
|---|---|---|---|---|---|---|---|---|---|
| | $\Delta$ Cov | PI-Width | Winkler | $\Delta$ Cov | PI-Width | Winkler | $\Delta$ Cov | PI-Width | Winkler |
| X / Y | −.004 | 2.69 | 1.27 | −.007 | 1.92 | 0.96 | −.010 | 1.54 | 0.81 |
| | (0.001) | (0.05) | (0.01) | (0.003) | (0.03) | (0.01) | (0.004) | (0.03) | (0.01) |
| X / Y / YHat | .011 | 2.75 | **0.96** | .012 | 2.16 | **0.77** | .009 | 1.82 | 0.68 |
| | (0.008) | (0.19) | (0.03) | (0.015) | (0.14) | (0.02) | (0.019) | (0.12) | (0.01) |
| X / Y / Model States | −.021 | **1.90** | 1.00 | −.035 | **1.43** | **0.77** | −.046 | **1.18** | **0.65** |
| | (0.003) | (0.04) | (0.01) | (0.004) | (0.03) | (0.01) | (0.005) | (0.02) | (0.00) |
| X / Y / YHat / Model States | −.018 | **1.91** | 1.00 | −.030 | **1.44** | 0.76 | −.041 | **1.19** | **0.65** |
| | (0.003) | (0.06) | (0.01) | (0.004) | (0.04) | (0.01) | (0.004) | (0.03) | (0.00) |

## Appendix D: Experiment Details

Each non-deterministic experiments, apart from experiments with HopCPT-G, are repeated with 12 different seeds. HopCPT-G experiments are repeated with 8 different seeds as the global models are computationally more demanding. For the latter experiments, we remove outlier runs where the training does not converge.

## Appendix E: Hyperparameter Search

We conducted a hyperparameter grid search for each model. In the case of HopCPT, we did this individually for each input
feature set. Each hyperparameter search was repeated with 3 seeds — the best average validation score was used as selection criteria. The validation score for the LSTM and CMAL is the NSE metric, for HopCPT, Bluecat, and kNN we followed (Auer et al., 2023) and used the smallest PI-Width at an epoch with $\Delta$ Cov $\leq 0$. To limit the number of grid search combinations for the LSTM and CMAL model, we split the hyperparameter into two sets which were tuned sequentially. The second set was trained given the result from the first set. Table E1 shows the parameters used in the hyperparameter search for the LSTM,
CMAL, HopCPT, and HopCPT-G models. For Bluecat we evaluated for $M = \{25, 50, 75, 100, 125, 150, 200\}$. For kNN we varied the share $k_s$ of samples which defines the $k$ parameter (i.e. $k = k_s *$ number of memory samples) and evaluated for $k_s = \{0.025, 0.05, 0.1, 0.15, 0.2, 0.25, 0.3, 0.35\}$.





Table E1. Parameters used in the hyperparameter search.

| Model | | Parameter | Values |
|---|---|---|---|
| CMAL | Set1 | hidden size | 60, 125, 250, 500 |
| | | output dropout | 0.4, 0.5 |
| | | target noise | 0.05, 0.1, 0.2 |
| | | # distrubtions | 1,3,5,10 |
| | Set2 | batch size | 256, 512 |
| | | learning rate | 0.0005, 0.0001, 0.001 |
| LSTM | Set1 | hidden size | 60, 125, 250, 500 |
| | | output dropout | 0.4, 0.5 |
| | | target noise | 0.05, 0.1, 0.2 |
| | Set2 | batch size | 256, 512 |
| | | learning rate | 0.0005, 0.0001, 0.001 |
| HopCPT | | learning rate | 0.001,0.001 |
| | | encode hidden layer | 1,2,3 |
| | | encode dropout | 0, 0.1 |
| | | temporal encoding | yes,no |
| HopCPT-G | | learning rate | 0.001,0.001 |
| | | encode hidden layer | 0,1,2,3 |
| | | encode dropout | 0, 0.1 |

## Appendix F: Point Prediction Metrics

Table F1 shows the point prediction performance of LSTM and CMAL for Experiments d. The performance for the PUB
prediction setting (Experiment II-b) is presented in Table F2. As in Klotz et al. (2022), CMAL outperforms the LSTM slightly.
This could imply an advantage for the interval predicting. However, this is justifiable, considering CMAL's greater volume of
training data as it does not need any calibration data.

## Appendix G: Extended Results

### G1  Experiment I-b - Additional Shift Examples

In Experiment I-b we show an example where a shift in the data highlights the advantages of the online mode for HopCPT.
Figure G1 and G2 show two additional examples, both from the same basin, at different months.





**Table F1.** Point prediction metrics for the experiments 1-5.1. NSE: Nash–Sutcliffe efficiency $(-\infty, 1]$; high values are better. MSE: Mean Squared Error; low values are better. KGE: Kling–Gupta efficiency $(-\infty, 1]$; high values are better. The variability of the metrics over the different basins and seeds is provided in the form of the standard deviation for the mean aggregation and in the form of the IQR for the median aggregation.

|  |  | LSTM | CMAL |
| --- | --- | --- | --- |
| NSE | Mean | 0.717 (0.297) | 0.716 (0.385) |
|  | Median | 0.754 (0.136) | 0.790 (0.157) |
| MSE | Mean | 2.363 (3.63) | 2.293 (5.025) |
|  | Median | 1.257 (1.955) | 0.996 (1.769) |
| KGE | Mean | 0.754 (0.254) | 0.733 (0.215) |
|  | Median | 0.817 (0.140) | 0.786 (0.168) |

**Table F2.** Point prediction metrics Experiment 5.2 (PUB). The values represent the average over all test basins of all splits. NSE: Nash–Sutcliffe efficiency $(-\infty, 1]$; high values are better. MSE: Mean Squared Error; low values are better. KGE: Kling–Gupta efficiency $(-\infty, 1]$; high values are better. The variability of the metrics over the different basins and seeds is provided in the form of the standard deviation for the mean aggregation and in the form of the IQR for the median aggregation.

|  |  | LSTM | CMAL |
| --- | --- | --- | --- |
| NSE | Mean | 0.444 (2.111) | 0.472 (2.394) |
|  | Median | 0.703 (0.243) | 0.690 (0.253) |
| MSE | Mean | 3.396 (6.603) | 3.362 (6.718) |
|  | Median | 1.365 (2.560) | 1.369 (2.545) |
| KGE | Mean | 0.496 (7.245) | 0.524 (0.591) |
|  | Median | 0.654 (0.326) | 0.638 (0.280) |





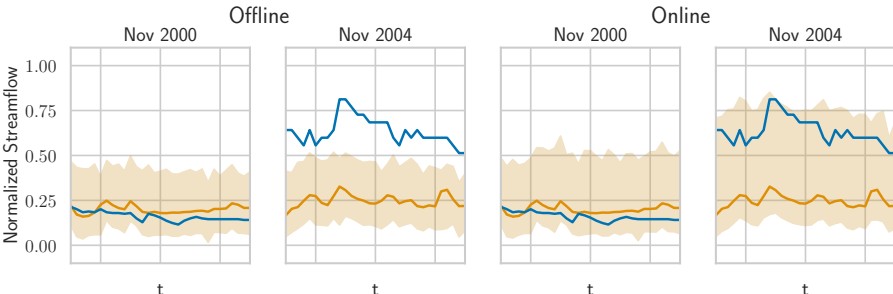

**Figure G1.** The real streamflow (gold), the prediction (blue), and HopCPT's prediction interval (light gold) for a basin (id: 12447390) in November 2000 and October 2004. The two plots one the left show HopCPT in the offline mode — the right two plots in the online mode. The online mode allows HopCPT to account for the distribution shift in November 2004.

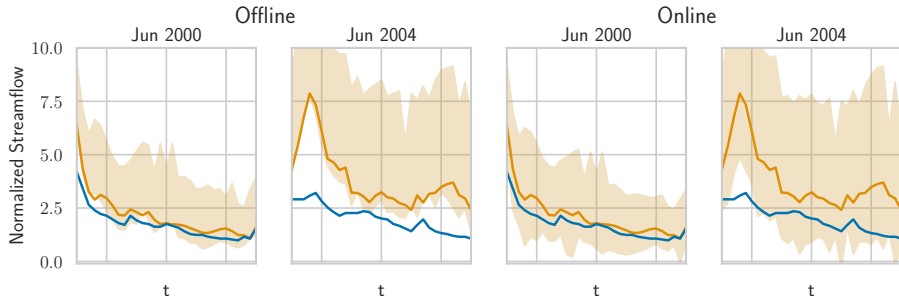

**Figure G2.** The real streamflow (gold), the prediction (blue), and HopCPT's prediction interval (light gold) for a basin (id: 12447390) in June 2000 and October 2004. The two plots one the left show HopCPT in the offline mode — the right two plots in the online mode. The online mode allows HopCPT to account for the distribution shift in June 2004.

### G2 Experiment II-b - PUB Results per Fold

Table G1 shows the results of the PUB experiment for the individual folds. The performance varies between the folds, however, the ranking of the different models is consistent over all folds.

### G3 Experiment III - Additional Peak Flow Evaluations

Figure G3 shows the peak flow results for HopCPT with different memory update frequencies (see Experiment I-b). The coverage difference between different update frequencies does not notably change when only peak flows are considered. Figure G4 presents the peak flow results for the PUB setting (see Experiment II-b). Similarly as in the "gauged" basin setting the PUB training improves the coverage of HopCPT notably. CMAL coverage is rather constant over the different peak flow shares.





**Table G1.** PUB performance over the individual folds for the miscoverage levels $\alpha = \{0.05, 0.10, 0.15\}$. The values represent the average over all test basins of the respective fold. Bold numbers correspond to the best result for the respective metric in the experiment (PI-Width and Winkler score). The values in parenthesis represent the standard deviation over the different seeds.

| | $\alpha$ | 0.05 | | | 0.10 | | | 0.15 | | |
|---|---|---|---|---|---|---|---|---|---|---|
| | | $\Delta$ Cov | PI-Width | Winkler | $\Delta$ Cov | PI-Width | Winkler | $\Delta$ Cov | PI-Width | Winkler |
| **Fold 1** | HopCPT-G | −.035 | **2.21** | 1.59 | −.048 | **1.62** | 1.17 | −.056 | **1.32** | 0.96 |
| | | (0.003) | (0.03) | (0.04) | (0.005) | (0.03) | (0.02) | (0.005) | (0.02) | (0.02) |
| | HopCPT-G (PUB Train) | .000 | 3.13 | 1.23 | −.008 | 2.34 | 0.97 | −.018 | 1.92 | 0.84 |
| | | (0.006) | (0.25) | (0.03) | (0.009) | (0.20) | (0.02) | (0.012) | (0.17) | (0.01) |
| | CMAL | −.097 | 2.37 | **1.10** | −.125 | 1.86 | **0.87** | −.140 | 1.57 | **0.76** |
| | | (0.030) | (0.15) | (0.04) | (0.039) | (0.12) | (0.03) | (0.043) | (0.11) | (0.02) |
| **Fold 2** | HopCPT-G | −.052 | **2.13** | 2.06 | −.064 | **1.52** | 1.43 | −.068 | **1.22** | 1.14 |
| | | (0.005) | (0.05) | (0.04) | (0.006) | (0.04) | (0.02) | (0.008) | (0.03) | (0.01) |
| | HopCPT-G (PUB Train) | −.020 | 2.75 | 1.48 | −.034 | 2.08 | 1.13 | −.049 | 1.71 | 0.96 |
| | | (0.011) | (0.24) | (0.10) | (0.015) | (0.17) | (0.05) | (0.015) | (0.14) | (0.03) |
| | CMAL | −.179 | 2.36 | **1.31** | −.218 | 1.85 | **1.00** | −.232 | 1.55 | **0.85** |
| | | (0.026) | (0.20) | (0.12) | (0.027) | (0.16) | (0.07) | (0.028) | (0.13) | (0.05) |
| **Fold 3** | HopCPT-G | −.050 | **1.95** | 1.39 | −.072 | **1.41** | 1.02 | −.089 | **1.13** | 0.84 |
| | | (0.004) | (0.04) | (0.02) | (0.006) | (0.03) | (0.01) | (0.008) | (0.03) | (0.01) |
| | HopCPT-G (PUB Train) | −.011 | 2.53 | 1.16 | −.020 | 1.92 | 0.88 | −.032 | 1.58 | 0.74 |
| | | (0.002) | (0.12) | (0.02) | (0.003) | (0.10) | (0.01) | (0.007) | (0.08) | (0.01) |
| | CMAL | −.121 | 1.98 | **0.90** | −.155 | 1.55 | **0.71** | −.170 | 1.30 | **0.62** |
| | | (0.019) | (0.14) | (0.06) | (0.019) | (0.11) | (0.04) | (0.018) | (0.10) | (0.03) |
| **Fold 4** | HopCPT-G | −.030 | **2.18** | 1.47 | −.042 | **1.58** | 1.08 | −.050 | **1.27** | 0.90 |
| | | (0.005) | (0.04) | (0.02) | (0.007) | (0.03) | (0.01) | (0.008) | (0.02) | (0.01) |
| | HopCPT-G (PUB Train) | −.002 | 2.74 | 1.18 | −.005 | 2.11 | 0.92 | −.008 | 1.76 | 0.78 |
| | | (0.008) | (0.23) | (0.06) | (0.013) | (0.18) | (0.03) | (0.015) | (0.15) | (0.02) |
| | CMAL | −.090 | 2.47 | **1.05** | −.119 | 1.94 | **0.82** | −.135 | 1.63 | **0.71** |
| | | (0.011) | (0.18) | (0.05) | (0.015) | (0.14) | (0.03) | (0.018) | (0.12) | (0.03) |
| **Fold 5** | HopCPT-G | −.079 | **2.34** | 2.59 | −.106 | **1.72** | 1.82 | −.119 | **1.39** | 1.46 |
| | | (0.002) | (0.04) | (0.05) | (0.004) | (0.02) | (0.02) | (0.005) | (0.02) | (0.01) |
| | HopCPT-G (PUB Train) | −.035 | 3.32 | 1.79 | −.057 | 2.51 | 1.41 | −.073 | 2.07 | 1.20 |
| | | (0.006) | (0.09) | (0.08) | (0.010) | (0.07) | (0.05) | (0.011) | (0.06) | (0.03) |
| | CMAL | −.128 | 2.92 | **1.58** | −.170 | 2.30 | **1.26** | −.189 | 1.94 | **1.09** |
| | | (0.032) | (0.27) | (0.13) | (0.040) | (0.20) | (0.10) | (0.044) | (0.17) | (0.09) |





| | $\alpha$ | 0.05 | | | 0.10 | | | 0.15 | | |
|---|---|---|---|---|---|---|---|---|---|---|
| | | $\Delta$ Cov | PI-Width | Winkler | $\Delta$ Cov | PI-Width | Winkler | $\Delta$ Cov | PI-Width | Winkler |
| **Fold 6** | HopCPT-G | −.050 | **2.25** | 1.58 | −.078 | **1.66** | 1.19 | −.093 | **1.34** | 1.00 |
| | | (0.005) | (0.03) | (0.03) | (0.006) | (0.03) | (0.02) | (0.007) | (0.03) | (0.01) |
| | HopCPT-G (PUB Train) | .004 | 3.18 | 1.28 | −.007 | 2.41 | 1.01 | −.020 | 1.99 | 0.87 |
| | | (0.011) | (0.32) | (0.04) | (0.016) | (0.24) | (0.03) | (0.018) | (0.19) | (0.02) |
| | CMAL | −.103 | 2.55 | **1.18** | −.143 | 2.00 | **0.93** | −.161 | 1.68 | **0.81** |
| | | (0.021) | (0.22) | (0.07) | (0.023) | (0.18) | (0.04) | (0.022) | (0.15) | (0.03) |
| **Fold 7** | HopCPT-G | −.050 | **2.15** | 1.77 | −.065 | **1.55** | 1.27 | −.072 | **1.25** | 1.04 |
| | | (0.006) | (0.04) | (0.04) | (0.006) | (0.03) | (0.02) | (0.008) | (0.02) | (0.01) |
| | HopCPT-G (PUB Train) | .000 | 3.14 | 1.34 | −.001 | 2.34 | 1.05 | −.001 | 1.91 | 0.90 |
| | | (0.011) | (0.33) | (0.08) | (0.014) | (0.23) | (0.04) | (0.014) | (0.18) | (0.03) |
| | CMAL | −.105 | 2.46 | **1.17** | −.142 | 1.92 | **0.93** | −.163 | 1.62 | **0.81** |
| | | (0.018) | (0.23) | (0.07) | (0.025) | (0.17) | (0.05) | (0.030) | (0.14) | (0.05) |
| **Fold 8** | HopCPT-G | −.046 | **2.15** | 1.66 | −.064 | **1.58** | 1.22 | −.072 | **1.28** | 1.01 |
| | | (0.004) | (0.04) | (0.03) | (0.005) | (0.03) | (0.02) | (0.005) | (0.02) | (0.01) |
| | HopCPT-G (PUB Train) | −.002 | 3.13 | 1.34 | −.010 | 2.38 | 1.04 | −.020 | 1.97 | 0.89 |
| | | (0.007) | (0.22) | (0.03) | (0.007) | (0.17) | (0.01) | (0.010) | (0.14) | (0.01) |
| | CMAL | −.123 | 2.38 | **1.18** | −.161 | 1.87 | **0.94** | −.180 | 1.57 | **0.82** |
| | | (0.024) | (0.09) | (0.03) | (0.024) | (0.08) | (0.03) | (0.023) | (0.07) | (0.03) |
| **Fold 9** | HopCPT-G | −.047 | **2.09** | 1.71 | −.060 | **1.53** | 1.23 | −.065 | **1.23** | 1.00 |
| | | (0.006) | (0.07) | (0.08) | (0.006) | (0.05) | (0.04) | (0.006) | (0.05) | (0.03) |
| | HopCPT-G (PUB Train) | .007 | 2.92 | 1.25 | .004 | 2.21 | 0.97 | −.002 | 1.82 | 0.83 |
| | | (0.008) | (0.15) | (0.04) | (0.017) | (0.12) | (0.02) | (0.023) | (0.10) | (0.02) |
| | CMAL | −.122 | 2.34 | **1.15** | −.162 | 1.82 | **0.90** | −.178 | 1.53 | **0.77** |
| | | (0.031) | (0.17) | (0.05) | (0.034) | (0.13) | (0.03) | (0.033) | (0.11) | (0.02) |
| **Fold 10** | HopCPT-G | −.079 | **2.10** | 1.96 | −.100 | **1.55** | 1.39 | −.109 | **1.26** | 1.13 |
| | | (0.007) | (0.07) | (0.08) | (0.010) | (0.06) | (0.04) | (0.012) | (0.05) | (0.03) |
| | HopCPT-G (PUB Train)) | −.020 | 3.18 | 1.42 | −.035 | 2.41 | 1.12 | −.047 | 1.99 | 0.96 |
| | | (0.011) | (0.18) | (0.03) | (0.012) | (0.13) | (0.02) | (0.014) | (0.11) | (0.01) |
| | CMAL | −.127 | 2.62 | **1.24** | −.157 | 2.06 | **0.96** | −.168 | 1.74 | **0.82** |
| | | (0.031) | (0.20) | (0.05) | (0.031) | (0.16) | (0.04) | (0.030) | (0.13) | (0.04) |

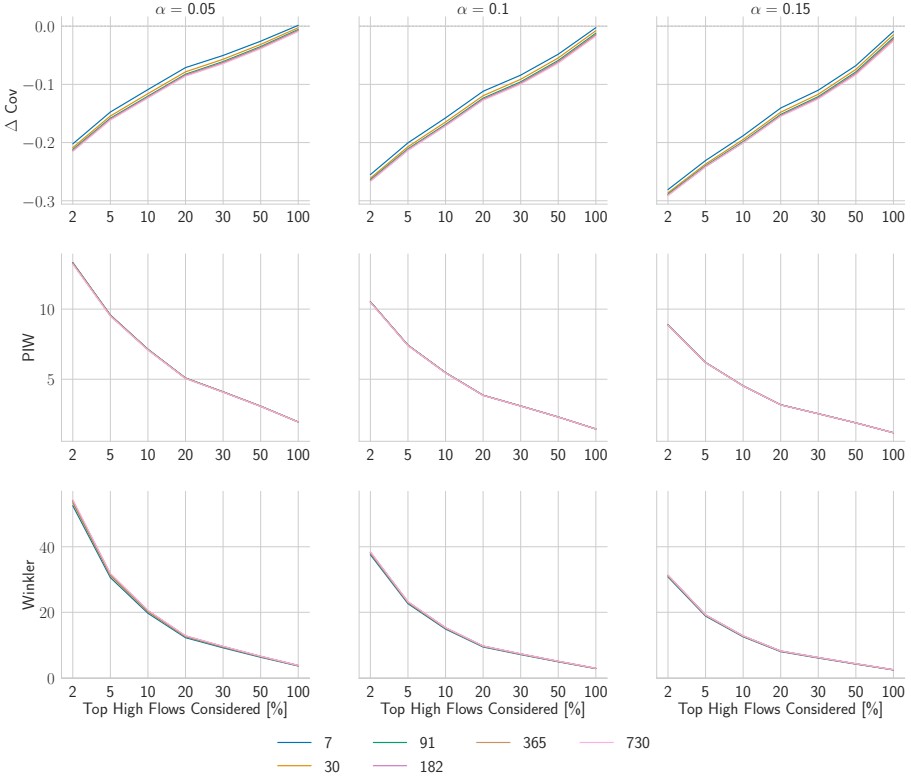

**Figure G3.** Evaluation metrics for high flows with a varying share of considered time steps. The different runs represent different update frequencies. (setting of Experiment I-b)

*Author contributions.* AA, DK, and MG designed all experiments. GN and FK checked the adequacy of the experiments and provided feedback. AA conducted all experiments, the results were analyzed by AA, DK, and MG. GN, DK, AA, and MG designed the structure of the exposition All authors contributed to the writing process with AA, DK, and MG doing the majority of the writing. AA created all figures. DK and SH supervised the manuscript.

*Competing interests.* The authors declare that they have no conflict of interest.

*Acknowledgements.* The ELLIS Unit Linz, the LIT AI Lab, the Institute for Machine Learning, are supported by the Federal State Upper Austria. We thank the projects Medical Cognitive Computing Center (MC3), INCONTROL-RL (FFG-881064), PRIMAL (FFG-873979), S3AI (FFG-872172), DL for GranularFlow (FFG-871302), EPILEPSIA (FFG-892171), AIRI FG 9-N (FWF-36284, FWF-36235), AI4GreenHeatingGrids (FFG- 899943), INTEGRATE (FFG-892418), ELISE (H2020-ICT-2019-3 ID: 951847), Stars4Waters (HORIZON-CL6-2021-CLIMATE-01-01). We thank Audi.JKU Deep Learning Center, TGW LOGISTICS GROUP GMBH, Silicon Austria Labs (SAL), FILL Gesellschaft

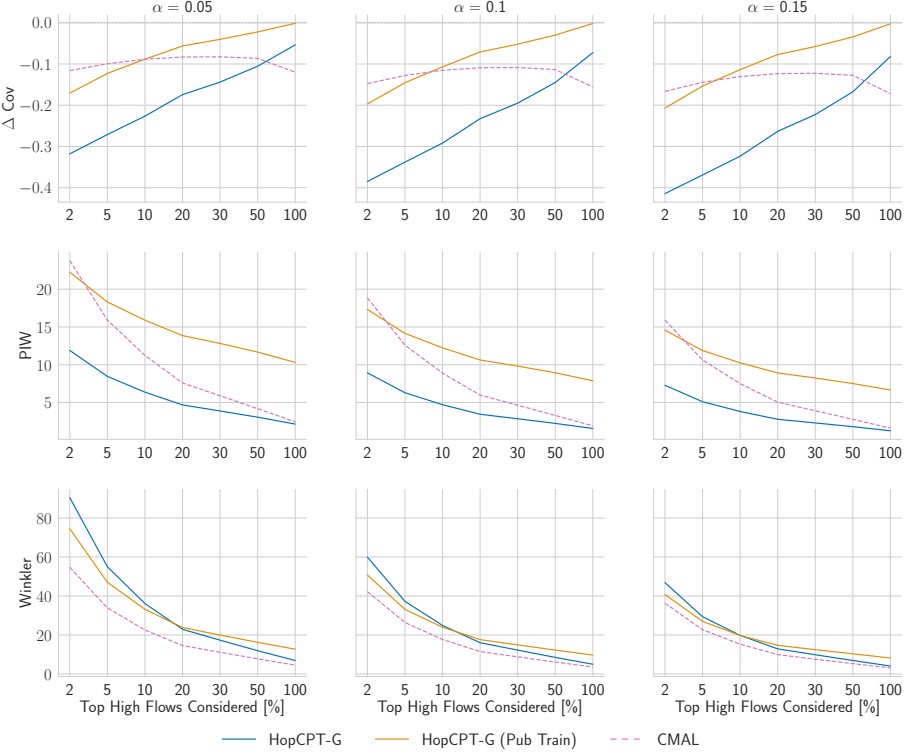

**Figure G4.** Evaluation metrics for high flows with varying share of considered time steps in the "ungagued basins setting" (setting of Experiment II-b)

mbH, Anyline GmbH, Google, ZF Friedrichshafen AG, Robert Bosch GmbH, UCB Biopharma SRL, Merck Healthcare KGaA, Verbund AG, GLS (Univ. Waterloo), Software Competence Center Hagenberg GmbH, Borealis AG, TÜV Austria, Frauscher Sensonic, TRUMPF and the NVIDIA Corporation. We acknowledge EuroHPC Joint Undertaking for awarding us access to LUMI at CSC, Finland. Use as many instances of the pattern LUMI at CSC, Finland as the number of systems awarded via EuroHPC.





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
