# Peer review of "A data-centric perspective on the information needed for hydrological uncertainty predictions"

_Hydrology and Earth System Sciences, 2024_

## Author Response (AR1)

**Point-to-Point Response**

A data-centric perspective on the information needed for hydrological uncertainty predictions

**We thank the reviewers for their thoughtful and constructive reviews. We appreciate their suggestions and questions. Below, we address each of their points.**
(Reviewer comments are formatted in italic - our answers are formatted in boldface - references to changes in the manuscript are underlined)

**Review 1:**

*This is a very interesting contribution, which has the potential of setting a new standard for error modelling in hydrology. I only have a few suggestions how to improve the presentation (see below). Apart from that, I have a conceptual question: The authors suggest to split the data into training data for the LSTM model and calibration data for the error model. Apart from the fact that I find this terminology a bit strange and potentially confusing in this context, I wonder whether it could be possible to train the parameters of both the model and the error model jointly? In traditional, likelihood-based probabilistic modeling this is common practice and just called "calibration".*

**We are aware of the complications surrounding the term "calibration" here. The problem is that the terminology overlaps between hydrology and probability: In probabilistic literature, calibration refers to the agreement between the observed frequencies of events and the probabilities assigned to those events. Hence, in the conformal prediction literature the "calibration data" is specifically reserved for the data which is not used as training data for the base model (and therefore does not suffer from overfitting bias). We tried to highlight this with footnote 1 on page 5, but we agree that this is not stringent enough. We made this aspect more clear in the revised manuscript. (lines 141-143 + footnote)**

**That said, a "joint training procedure" of a prediction model and an error model is generally not possible in the framework of conformal prediction. This is because the prediction model can arbitrarily overfit on the training data and if one would estimate error bands or probabilities on top of that we would get overconfident uncertainty estimates (i.e., very thin bars). However, this is not a specific limitation of conformal prediction as observations on the training data are in general distorted by overfitting. We discuss this point in the revised manuscript. (line 143 + footnote)**

*Another interesting question could be: How do the error bars respond to changes in input variables? E.g. what happens when I increase the rain while keeping all the other variables fixed? This would*

*not only be a sanity check of the data-driven model (do the errors respond in a reasonable way?) but it could also lead to a better understanding of what drives the errors.*

**We agree that experiments with input perturbation would be very interesting and could potentially give an even deeper understanding of how such models work. However, we think a thorough analysis regarding this line of experiments is out of the scope of this paper as we are mainly interested in which data is necessary --- both in the spatial and temporal dimensions --- to generate reliable uncertainty estimates. However, since we do in principle endorse the idea we added it to our discussion on potential future work (lines 393-394), with an appropriate reference to the reviewer in the acknowledgments.**

*Regarding the presentation, I think that a bit more details and context would make the manuscript more accessible, especially for people who have never heard of these techniques before. For instance, in eq. (1), one is left to wonder what the dimensions of the objects are. Furthermore, it is not clear how the training works until one reads on to eq. (4).*

**Thank you for pointing out the missing variable dimension in Eq 1. The dimensions are as follows: $m(z_i), m(z_t) \ldots R^{d'}$; $z_i, z_t \ldots R^d$;
$W_q, W_k \ldots R^{\{d' x d'\}}$ where d' is the hidden dimension of the error model and d is the dimension of the input to the error model. We added this information in the revised manuscript. (line 147-154)
Regarding the clarification of the training process, we now highlight in the revised manuscript at the beginning of Section 2.2.1 that the model is optimized with an MSE-like loss function and gradient descent and add more detailed information in the Appendix.
(lines 156-158 + Appendix C)**

*It would also be interesting to relate the MHN approach to the very popular attention mechanism of transformers and provide a bit more context as to why this is a reasonable approach.*

**MHNs are closely related to Transformers (Vaswani et al., 2017) by their association/attention mechanism. In fact, Ramsauer et al. (2021) show that the Transformer attention is a special case of the MHN association, at which we arrive when the queries and keys are mapped in a certain way and given a specific inverse softmax temperature. However, the framework of MHN highlights the associative memory mechanism, as HopCPT directly ingests encoded observations. This associative mechanism allows HopCPT to retrieve error information from situations that are similar to the given one. In the global variant (HopCPT-G) this even allows to draw from error information of different basins.
MHNs further have the advantageous properties of exponential storage capacity and one-step update retrieval (Ramsauer et al., 2021).
We added this information to the revised manuscript. (lines 173-174 + Appendix D)**

*When it comes to the presentation of the results, some typical hydrographs with error bands would offer some visual support to the summary tables.*

**We agree that hydrographs would help to support the results and understand the model behavior. As there are a multitude of potential hydrographs, given the dataset size and the different models, we select a specific setting to showcase the predictions of HopCPT and add the respective hydrograph to Section 2.2 where we introduce the model. (Figure 2)**

***References***

Ramsauer, H., Schafl, B., Lehner, J., Seidl, P., Widrich, M., Gruber, L., Holzleitner, M., Pavlovic, M., ´ Sandve, G. K., Greiff, V., Kreil, D., Kopp, M., Klambauer, G., Brandstetter, J., and Hochreiter, S. Hopfield networks is all you need. In 9th International Conference on Learning Representations (ICLR), 2021.

Vaswani, A., Shazeer, N., Parmar, N., Uszkoreit, J., Jones, L., Gomez, A. N., Kaiser, L., and Polosukhin, I. Attention is all you need. In Guyon, I., Luxburg, U. V., Bengio, S., Wallach, H., Fergus, R., Vishwanathan, S., and Garnett, R. (eds.), Advances in Neural Information Processing Systems 30, pp. 5998–6008. Curran Associates, Inc., 2017

**Review 2**

*The manuscript proposes to use the framework of Conformal Prediction to produce uncertainty estimates for data-driven predictive models. Different variants of the proposed approach are tested and thoroughly discussed to investigate the role of input data both in time and space. This is a very interesting, novel, and well-designed research that can become very valuable to the hydrologic community where the use of ML models for hydrologic forecasting is continuously growing.*
*To maximize the impact of this paper, I would suggest the authors to consider addressing the following two major points:*

*1) the current structure of the introduction might represent an obstacle to reaching a wider community as the motivations of the study - albeit clear - are basically synthesized in the very first sentence of the paper; then, the authors jump directly into the proposed approach waiting till line 50+ to discuss other existing approaches for uncertainty estimation in hydrology. I believe the paper would benefit from a less abrupt intro that expands the motivations of this paper and summarizes existing methods to better clarify, especially for a non-ML expert readership, the positioning of this work with respect to the existing literature.*

**We agree that an extension of the first part of the introduction (i.e., the general motivation of the paper) would improve the accessibility of the manuscript. We rephrased the start of the manuscript as follows:**
**"Uncertainty estimates are the basis for actionable predictions (e.g., Krzysztofowicz, 2001;**

Beven, 2016a). For example, the decision to provide a flood warning does not only depend on the size of a given peak but also on how likely one thinks the event will occur. For a modeler it is therefore natural to ask what information is necessary or useful to provide high-quality uncertainty estimates. In this contribution, we take a data-centric perspective to investigate this question. Specifically, we analyze how temporal (recency) and spatial (local vs. global) information impacts the quality of uncertainty estimates. The focus on temporal and spatial information is motivated by findings in existing literature, indicating that information from different regions could compensate for a lack in temporal richness (Bertola et al., 2023; Nearing et al., 2024; Kratzert et al., 2024)" (lines 1-16)

To highlight related uncertainty models for hydrology earlier in the introduction, we will reorder the paragraphs "Uncertainty estimation in hydrology" and "Trading space for time in hydrological modeling" in the revised manuscript. (line 39*ff.* and line 49*ff.*)

*2) building on the previous point and moving from the presentation to the research experiments, at the end of the paper I had the impression that the experimental settings could be improved by considering some form of benchmarking vs existing traditional methods for uncertainty estimation. As the authors report in the introduction (lines 51-59), several methods have been used in the hydrologic community. Showing how the proposed approach performs against these traditional methods could further reinforce the numerical analysis and ultimately convince also non-ML experts about the value of the proposed method (as happened with the paper by Mai et al. 2022 co-authored by some of the authors).*

In this work, we predominantly use HopCPT to analyze which aspects of the data are important to generate good uncertainty estimates. That is, our goal is not to benchmark the quality of uncertainty predictions. Benchmarking for uncertainty predictions is an interesting question in its own right. Nonetheless, we agree that a general comparison of the model performance is helpful, as HopCPT can also be used to conduct uncertainty estimation in practice. To address this, we have included a performance comparison to three other models in Appendix~A. Therein, we compare (1) with an optimized version of Bluecat (Koutsoyiannis and Montanari, 2022), with a detailed analysis of our modifications and their impact on the approach's performance in Appendix~B; (2) to the state-of-the-art method CMAL (Klotz et al., 2022); and (3) to a k-nearest-neighbor (kNN) approach. While we originally conceived this kNN comparison as a simplified version of HopCPT (to highlight the need for learned feature representations), we later found that this approach was in fact proposed as a hydrological uncertainty estimation method by Wani et al. (2017). We added this missing reference in the revised manuscript.

We also agree with the reviewer that the aspect of model comparison is only minimally referenced in the main part of the manuscript. Hence, we highlight the model comparison and the respective appendix more prominently in the revised manuscript. (lines 251-253)

*Moreover, another point worth discussing could be the following: the authors suggest implementing the proposed approach by training the prediction model on different data than those used for calibrating HopCPT. Can you elaborate on the resulting data requirements, considering that often ML applications in the hydrologic domain face important challenges related to limited data availability? This research uses the CAMELS dataset, how difficult could be to apply this in data-scarce regions? In such contexts, is there any room for potentially focusing on one phase only and "importing" the other from a completely different region, i.e. I train the prediction model in the data-scarce region an use HopCPT calibrated on CAMELS in PUB mode (or vice versa)?*

**HopCPT requires data that is not part of the training data of the prediction model. This property is very important to us because the prediction model can arbitrarily overfit on the training data and if one were to estimate error bands or probabilities on top of that, we would get overconfident uncertainty estimates (i.e., very thin bars). We would also like to point out that this is not a specific limitation of conformal prediction or HopCPT as such, since predictions on the training data are in general distorted because of overfitting phenomena. That said, we do agree with the reviewer that it is worth discussing these technicalities in more detail. (lines 141-143 + footnote)**

**Further, as the reviewer rightly suggests, an idea for data-sparse regions could be that training is done together with other data, e.g., CAMELS. One can view the PUB case as the extreme example of this, where no data for the predicted region is available. If there is at least some data for the predicted region you could either add it to the training data of the prediction model or add it to the data used by HopCPT. While the former might be the better choice if you predominantly care about the prediction quality the latter would be favorable in case a better uncertainty estimation quality is the main objective. Splitting the data could be a good trade-off.**
**We added a note about the possibility of information transfer to data-sparse basins in the revised manuscript (lines 201-202)**

*MINOR COMMENTS:*

*- at line 30 the acronym PUB is not defined*

*- at line 148 there is a repetition of "that"*

**We thank the reviewer for pointing that out. We added the definition for PUB (line 34) and removed the repetition of "that" in the revised manuscript.**

*References*

Bertola, M., Blöschl, G., Bohac, M. et al.: Megafloods in Europe can be anticipated from observations in hydrologically similar catchments. Nat. Geosci. 16, 982–988 (2023).

Beven, K.: Facets of uncertainty: epistemic uncertainty, non-stationarity, likelihood, hypothesis testing, and communication, Hydrological Sciences Journal, 61, 1652–1665, (2016a)

Klotz, D., Kratzert, F., Gauch, M., Keefe Sampson, A., Brandstetter, J., Klambauer, G., Hochreiter, S., and Nearing, G.: Uncertainty estimation with deep learning for rainfall–runoff modeling, Hydrol. Earth Syst. Sci., 26, 1673–1693, (2022)

Koutsoyiannis, D., & Montanari, A.: Bluecat: A local uncertainty estimator for deterministic simulations and predictions. Water Resources Research, 58, (2022)

Kratzert, F., Gauch, M., Klotz, D., and Nearing, G.: HESS Opinions: Never train an LSTM on a single basin, Hydrol. Earth Syst. Sci. Discuss. [preprint], (2024)

Krzysztofowicz, R:. The case for probabilistic forecasting in hydrology. Journal of hydrology, 249(1-4), 2-9, (2001)

Nearing, G., Cohen, D., Dube, V. et al.: Global prediction of extreme floods in ungauged watersheds. Nature 627, 559–563 (2024).

Wani, O., Beckers, J. V. L., Weerts, A. H., and Solomatine, D. P.: Residual uncertainty estimation using instance-based learning with applications to hydrologic forecasting, Hydrol. Earth Syst. Sci., 21, 4021–4036, (2017).